# Structural basis of siRNA recognition by TRBP double-stranded RNA binding domains

Gregoire Masliah[1], Christophe Maris[1], Sebastian LB König[2], Maxim Yulikov[3], Florian Aeschimann[4], Anna L Malinowska[5], Julie Mabille[4], Jan Weiler[4], Andrea Holla[2], Juerg Hunziker[4], Nicole Meisner-Kober[4], Benjamin Schuler[2], Gunnar Jeschke[3] & Frederic H-T Allain[1],*

## Abstract

The accurate cleavage of pre-micro(mi)RNAs by Dicer and mi/siRNA guide strand selection are important steps in forming the RNA-induced silencing complex (RISC). The role of Dicer binding partner TRBP in these processes remains poorly understood. Here, we solved the solution structure of the two N-terminal dsRNA binding domains (dsRBDs) of TRBP in complex with a functionally asymmetric siRNA using NMR, EPR, and single-molecule spectroscopy. We find that siRNA recognition by the dsRBDs is not sequence-specific but rather depends on the RNA shape. The two dsRBDs can swap their binding sites, giving rise to two equally populated, pseudo-symmetrical complexes, showing that TRBP is not a primary sensor of siRNA asymmetry. Using our structure to model a Dicer-TRBP-siRNA ternary complex, we show that TRBP's dsRBDs and Dicer's RNase III domains bind a canonical 19 base pair siRNA on opposite sides, supporting a mechanism whereby TRBP influences Dicer-mediated cleavage accuracy by binding the dsRNA region of the pre-miRNA during Dicer cleavage.

**Keywords** Dicer; NMR; single-molecule FRET; siRNA; TRBP
**Subject Categories** RNA Biology; Structural Biology
**The EMBO Journal (2018) 37: e97089**

## Introduction

RNA interference (RNAi) is a control mechanism of gene expression relying on small non-coding RNAs (Mello & Conte, 2004; Carthew & Sontheimer, 2009). One hallmark of the RNAi pathway is the RNA-induced silencing complex (RISC), a ribonucleoprotein (RNP) comprising the protein Argonaute (Ago) along with a single-stranded RNA of 21–23 nucleotides called microRNA (miRNA) or short interfering RNA (siRNA) (Sontheimer, 2005). Gene silencing is brought about by the formation of Watson–Crick interactions between the miRNA component of the RISC and cognate mRNAs (Sontheimer, 2005).

miRNA biogenesis is initiated in the nucleus with RNA polymerase II transcribing the primary miRNA (pri-miRNA), a precursor containing one or several stem-loop elements flanked by single-stranded RNA regions (Bartel, 2004; Kim, 2005a,b). The pri-miRNA is processed by the microprocessor, a complex containing the ribonuclease III (RNase III) enzyme Drosha and the RNA-binding protein DGCR8, resulting in the release of the pre-miRNA (Denli et al, 2004; Gregory et al, 2004; Herbert et al, 2016; Kwon et al, 2016). Pre-miRNA secondary structure typically consists of a hairpin of ~70 nucleotides with irregularities such as bulges and internal loops, and a characteristic two-nucleotide overhang at its 3′ end. Pre-miRNA is exported through the exportin 5 complex into the cytoplasm (Lund et al, 2004; Kim et al, 2009), where the RNase III enzyme Dicer excises its apical loop, thereby producing a mature miRNA duplex displaying 2-nucleotide overhangs at both 3′ ends (Bernstein et al, 2001; Hutvágner et al, 2001). Following the transfer of the miRNA duplex to a member of the Argonaute family, one of the two RNA strands—the "passenger"—is removed from Ago, while the other strand—the "guide"—is retained in the mature RISC particle, a process known as "strand sorting" or "strand selection" (Sontheimer, 2005; Tomari & Zamore, 2005; Kawamata et al, 2009).

The molecular mechanisms governing strand selection are not yet completely understood. Structural studies of Ago in complex with various RNAs have shown that the 5′ and 3′ ends of the miRNA guide strand are bound to the MID and PAZ domains of Ago (Lingel et al, 2003, 2004; Song et al, 2003; Ma et al, 2004, 2005; Frank et al, 2010). When the miRNA duplex is loaded onto Ago, only the guide strand is retained (Matranga et al, 2005; Rand et al, 2005; Leuschner et al, 2006; Kawamata et al, 2009). The identity of the guide strand is therefore determined by the

1 Institute of Molecular Biology and Biophysics, ETH Zürich, Zürich, Switzerland
2 Department of Biochemistry, University of Zürich, Zürich, Switzerland
3 Laboratory of Physical Chemistry, ETH Zürich, Zürich, Switzerland
4 Novartis Institutes for Biomedical Research, Basel, Switzerland
5 Institute of Pharmaceutical Sciences, Department of Chemistry and Applied Biosciences, ETH Zürich, Zürich, Switzerland
*Corresponding author. Tel: +41 44 633 39 40; E-mail: allain@mol.biol.ethz.ch

orientation of the miRNA duplex and its interaction with the PAZ and MID domains of Ago. Several factors seem to play a role therein, such as the recognition of the nucleobase at the 5′ end of the guide strand by the MID domain of Ago (favoring a U or an A) (Frank *et al*, 2010; Suzuki *et al*, 2015) or the differential thermodynamic stability of the ends of the miRNA duplex (Khvorova *et al*, 2003; Schwarz *et al*, 2003). In flies, the heterodimer Dicer-2/R2D2 senses siRNA asymmetry and participates in RISC loading (Tomari *et al*, 2004). It has been reported, however, that Dcr2/R2D2 involvement in that process is not absolutely required (Nishida *et al*, 2013). In humans, Dicer has been shown to be dispensable for RISC loading (Betancur & Tomari, 2012) while TRBP alone could recognize siRNA asymmetrically (Gredell *et al*, 2010). Other reports suggest that strand selection is rather the result of a subtle interplay of the asymmetry rule, the identity of the 5′ nucleotides of the miRNA duplex, and the involvement of Dicer, TRBP, and Ago as multiple sensors (Noland *et al*, 2011; Noland & Doudna, 2013). In the same vein, several reports suggest that TRBP affects the processing of a particular set of premiRNA by shifting the cleavage site of Dicer by one nucleotide, causing the inversion of the guide/passenger strands for a subset of these pre-miRNAs (Fukunaga *et al*, 2012; Lee & Doudna, 2012; Kim *et al*, 2014; Wilson *et al*, 2015).

TRBP was first identified as a protein facilitating HIV infection (Gatignol *et al*, 1991). It is an RNA-binding protein of 39 kDa, which associates with Dicer (Haase *et al*, 2005), influences the precision of pre-miRNA cleavage (Kim *et al*, 2014; Wilson *et al*, 2015), and helps recruiting Ago (Chendrimada *et al*, 2005). TRBP homologs include Loquacious and R2D2 in *Drosophila melanogaster*, RDE-4 in *Caenorhabditis elegans*, DRB1-3,5 in *Arabidopsis thaliana*, Xlrbpa in *Xenopus laevis* (Eckmann & Jantsch, 1997), and the PKR activator (PACT) protein in mammals (Peters *et al*, 2001). TRBP and its homologs share the same domain architecture, with three consecutive dsRNA binding domains (dsRBDs) (St Johnston *et al*, 1992; Masliah *et al*, 2013) separated by linkers of various lengths. The affinity of TRBP for dsRNA is essentially conferred by the two N-terminal dsRBDs (Yamashita *et al*, 2011; Takahashi *et al*, 2013), whereas the third dsRBD does not bind dsRNA but interacts with Dicer (Daniels & Gatignol, 2012; Wilson *et al*, 2015). The tertiary structures of the first (dsRBD1) and second (dsRBD2) domains in their free form (Yamashita *et al*, 2011), of dsRBD2 in complex with RNA (Yang *et al*, 2010), and of the third dsRBD (dsRBD3) in complex with a Dicer fragment (Wilson *et al*, 2015) have been elucidated.

To gain deeper insight into the function of TRBP in RNAi processing, we solved the structure of its two N-terminal dsRBDs (dsRBD12) in complex with a highly asymmetric siRNA (EL86) using NMR in conjunction with EPR and single-molecule FRET. We observe that EL86 binds dsRBD12 in two different and opposite orientations, in equal proportions, indicating that EL86 asymmetry does not influence TRBP binding significantly. Our structures show that dsRBD12 covers one side of EL86 along its whole length, leaving the other face potentially accessible to other protein factors. We show experimentally that dsRBD12 does not interfere with pre-miRNA cleavage by Dicer, suggesting the possible existence of a ternary complex TRBP-Dicer-pre-miRNA during the cleavage step.

# Results

## Individual TRBP's dsRBDs bind EL86 in multiple registers

Protein fragments containing human TRBP (UniProtKB Q15633) dsRBD1 or dsRBD2 were titrated with EL86, a potent siRNA of 19 base pairs with characteristic two-nucleotide 3′-overhangs (Stalder *et al*, 2013; Fig 1A). Formation of each dsRBD-RNA complex was followed by NMR spectroscopy and resulted in large chemical shift changes (Fig EV1A). Mapping these chemical shift perturbations onto the crystal structure of a dsRBD2-dsRNA complex (Ryter & Schultz, 1998) or on the dsRBDs' primary structures reveals three clusters corresponding to the three regions composing the canonical RNA binding surface of dsRBDs (helix α1, loop 2, N-terminal tip of helix α2) (Fig EV1B). We conclude from these observations that TRBP's individual domains dsRBD1 and dsRBD2 bind EL86 in a canonical fashion, that is, by interacting with two consecutive minor grooves of an A-form RNA helix (Ryter & Schultz, 1998; Masliah *et al*, 2013).

Next, we measured protein–RNA intermolecular NOEs in the dsRBD1-EL86 and dsRBD2-EL86 complexes using 3D $^{13}$C-edited filtered NOESY experiments to assess whether the two domains bind EL86 sequence specifically. Interestingly, a large number of intermolecular NOEs involving dsRBD residues located at the RNA binding surface were observed (Appendix Fig S1A and B). Unexpectedly, residues Ala57 (dsRBD1) and Ala187 (dsRBD2) (all sequence numbers correspond to TRBP wild-type numbering, Fig EV1C), which are located at equivalent positions in the β1–β2 loop of each domain, have intermolecular NOEs with at least eight consecutive residues located at the 3′ end of each EL86 strand (Fig 1B). In case of single-register binding, Ala57 and Ala187 are expected to have a maximum of two intermolecular NOEs, that is, with two consecutive nucleotides. Therefore, we conclude from this multiplicity of NOEs that TRBP's individual domains do not bind EL86 sequence specifically but rather in multiple registers, unlike the dsRBDs of ADAR2 or NF90, which have been shown to bind dsRNA in single registers (Stefl *et al*, 2010; Jayachandran *et al*, 2016).

## Tertiary structures of individual dsRBD1 and dsRBD2

Tertiary structures of TRBP's single domains dsRBD1 and dsRBD2 in the RNA-bound state were determined by NMR spectroscopy. Ensembles of 20 structures with precisions of 0.35 ± 0.11 Å and 0.22 ± 0.05 Å were obtained for dsRBD1 and dsRBD2, respectively (Appendix Table S1). As expected, both domains adopt the canonical dsRBD fold characterized by an αββββα topology. Interestingly, dsRBD1 presents a short additional α-helix at its N-terminus (referred as helix α0 hereafter), which folds back on the cleft between helices α1 and α2 (Fig 1C). Our structure shows that helix α0 is stabilized by several hydrophobic interactions involving side-chains from helix α0 (Ile19, Met22, and Leu23), helix α1 (Leu34 and Tyr38), and helix α2 (Leu92, Leu95, and Lys96). A similar extension was reported for the third dsRBD of ADAR 1 where it plays a role in cellular localization (Barraud *et al*, 2014).

Aside from helix α0, the structures of dsRBD1 and dsRBD2 are very similar, with a backbone r.m.s.d. of 1.22 ± 0.07 Å. In addition, our NMR structures superimpose very well with the crystal structures of dsRBD1 in the free state (Yamashita *et al*, 2011) and

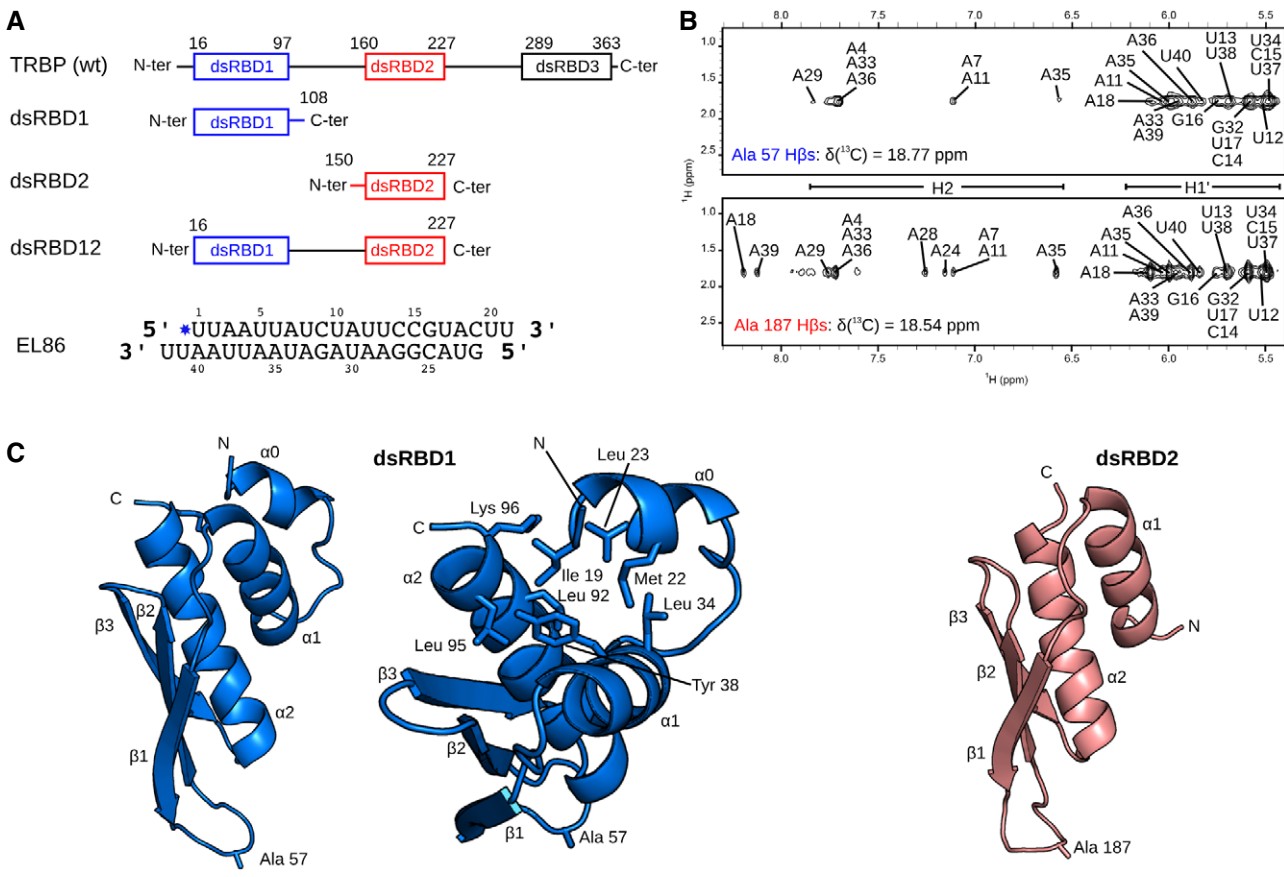

**Figure 1.  TRBP domains organization, inter molecular NOEs and three dimensional structures of dsRBD1 and dsRBD2.**

A  (Top) Domain architecture of full-length TRBP. Domain boundaries are indicated above each domain. (Middle) The following TRBP fragments were used in this study: dsRBD1 and dsRBD2, which contain a single dsRBD, dsRBD12, which contains both dsRBD1 and dsRBD2 separated by the native inter-domain linker. (Bottom) Nucleotide sequence and secondary structure of the siRNA EL86.

B  Intermolecular NOEs between EL86 H1′ protons and Ala57 (dsRBD1, upper) or Ala187 (dsRBD2, lower).

C  Solution structures of the EL86-bound forms of dsRBD1 and dsRBD2 determined by NMR spectroscopy. dsRBD1 and dsRBD2 are colored in blue and red, respectively. The N-terminal α-helix extension found in dsRBD1 is denoted as α0. Residues Ala57 and Ala187 whose intermolecular NOEs are shown in (B), and residues interacting with helix α0 protons are represented as sticks.

---

dsRBD2 bound to dsRNA (Yang *et al*, 2010), yielding backbone r.m.s.d. of 0.82 ± 0.04 Å and 0.85 ± 0.05 Å, respectively. We then compared the NMR fingerprints of dsRBD1, dsRBD2, and dsRBD12 in the presence of EL86 (Fig EV2). We do not observe any significant shift in peak positions and conclude therefore that the structures of dsRBD1 and dsRBD2 do not change significantly when they are expressed in tandem as within dsRBD12, as previously reported (Benoit *et al*, 2013; Wilson *et al*, 2015).

**In the dsRBD12-EL86 complex, dsRBD2 binds in two symmetric orientations**

Next, we studied siRNA binding by the N-terminal half of TRBP, which contains dsRBD1 and dsRBD2 in tandem (Fig 1A). Intermolecular NOEs collected on the dsRBD12-EL86 complex reveal that both dsRBDs interact with EL86, using the same binding surface as the isolated domains (Appendix Fig S1A–C). Furthermore, the absence of intermolecular NOEs from the inter-domain linker indicates that the linker does not interact with EL86, in agreement

with a study showing that TRBP's linker does not interact with pri-miR-155 (Benoit *et al*, 2013).

Within the dsRBD12-EL86 complex, the only protein side chain having unambiguous intermolecular NOEs is Leu175, which is located in the dsRBD2's loop connecting helix α1 to strand β1 (Figs 1C and 2B). Interestingly, the Leu175 methyl group Hδ1 has only four intermolecular NOEs with EL86 ribose H1′, whereas at least 10 peaks were observed in the dsRBD2-EL86 complex (Fig 2A). This suggests that dsRBD2 binds in fewer registers in the dsRBD12-EL86 complex than in the dsRBD2-EL86 complex. These four intermolecular NOE peaks could be unambiguously assigned to EL86 residues U2, A3, U23, and A24 (Fig 2A). U2-A3 and U23-A24 sit on opposite EL86 ends, separated by ~50 Å. They are thus too far away to be simultaneously contacted by dsRBD2. We conclude therefore that dsRBD2 binds EL86 predominantly in two different registers, in which Leu175 is positioned either at U2-A3 or U23-A24 (Fig 2B).

To validate this interpretation, we measured five pairwise distance distributions between dsRBD2 and EL86 in the dsRBD12-EL86

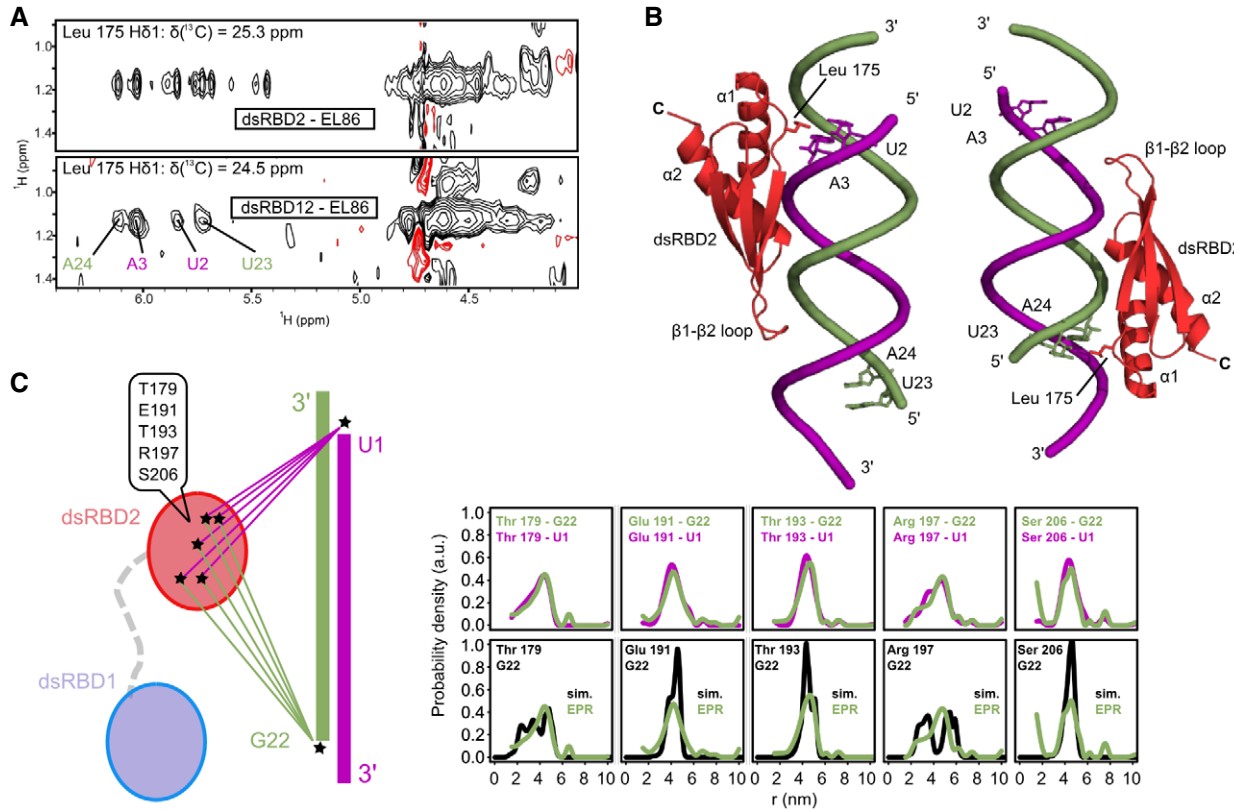

**Figure 2. DsRBD2 positioning on EL86 in RNA-bound dsRBD12.**

A  Selected regions of 3D $^{13}$C-edited filtered NOESY spectra collected on the dsRBD2-EL86 (top) and dsRBD12-EL86 (bottom) complexes, showing intermolecular NOE cross-peaks between dsRBD2 Leu175 Hδ1s and EL86 ribose protons. Unambiguous assignments of EL86 H1′ are colored in blue (upper strand) and green (lower strand).

B  Structural models of the dsRBD2-EL86 complex showing the two orientations compatible with the intermolecular NOEs observed in the dsRBD12-EL86 complex.

C  dsRBD2-EL86 distances measured by EPR in the dsRBD12-EL86 complex. (Left, scheme) Five pairwise distance distributions were measured between the dsRBD2 β-sheet surface (Cys179, Cys191, Cys193, Cys197, or Cys206) and the 5′ termini (Ura1 or Gua22) of EL86. DsRBD1 (light blue) and dsRBD2 (red) are connected by the native flexible linker (gray dashes). Nitroxide spin labels are represented by asterisks. (Right, upper row) Superposition of experimental distance distributions between dsRBD2 and either EL86 Ura1 (magenta) or Gua22 (green) strands. (Right lower row) Superposition of back-calculated (black) and experimental (green) distance distributions between EL86 Gua22 and each of the five spin labels attachment sites on dsRBD2. Simulated distance distributions were calculated from the two models of dsRBD2-EL86 shown in (B).

complex, using EPR spectroscopy and site-directed spin labeling (SDSL) (Jeschke *et al*, 2006; Bordignon, 2012; López *et al*, 2012). Single nitroxide spin labels were introduced at five specific sites on the dsRBD2 β-sheet surface of single mutants (T179C, E191C, T193C, R197C, S206C), and at either 5′ end of EL86, yielding a total of 10 different distances (Fig 2C). All measured distances are between 20 and 50 Å, a distance range that would be expected for dsRBD2 bound to EL86, as the latter is roughly 70 Å in length. We observe virtually identical distance distributions, independent of whether EL86 is labeled on U1 or G22 (Fig 2C). This indicates that the dsRBD2 registers are characterized by identical locations relative to either of the two EL86 termini, which is in agreement with the pattern of intermolecular NOEs observed with Leu175 (Fig 2A and B).

Next, we modeled two canonical dsRBD2-EL86 complexes in which Leu175 was positioned between either U2-A3 or U23-A24 (Fig 2B). dsRBD2-EL86 pairwise distances were simulated using both models and compared with distance distributions determined by EPR (Fig 2C). Very good agreement is observed for the shapes and the means of E191C, T193C and S206C distributions (Fig 2C). Interestingly, the experimental distributions obtained

with T179C and R197C are significantly broader, with significant contributions from both short (2–3 nm) and long (3.5–5.5 nm) distances. As shown by the simulations, each contribution originates from a specific orientation of dsRBD2 on EL86, supporting further the presence of two binding sites revealed by intermolecular NOEs.

## DsRBD12 and EL86 form two symmetric, equally populated, complexes

Because of the high level of ambiguity of dsRBD1's intermolecular NOEs, dsRBD1 was positioned with respect to dsRBD2 with the help of residual dipolar couplings (RDCs). Forty-four peptide backbone {$^{15}$N,$^{1}$H} RDCs were collected on the dsRBD12-EL86 complex. Calculation of a Pearson's correlation factor (*Rp*) for dsRBD1 and dsRBD2 yielded values of 0.93 and 0.95, respectively, demonstrating that the RDC dataset is in good agreement with the NMR structures of the individual domains. Interestingly, the two domains have the same alignment tensors, with magnitudes of 10.3 ± 0.5 Hz and 10.8 ± 0.4 Hz, and rhombicities of 0.25 ± 0.05 and 0.29 ± 0.05 for

dsRBD1 and dsRBD2, respectively (Appendix Table S2). This indicates that the relative orientation of dsRBD1 with respect to dsRBD2 is well defined and can be determined by analyzing the set of RDCs within a single coordinate frame common to both domains.

Fourteen models of the dsRBD12-EL86 complex were built, in which dsRBD2 was fixed at one of the two positions determined previously, whereas dsRBD1 was positioned in any remaining accessible register. The agreement of each model with the full RDC dataset was evaluated using a single alignment tensor for both dsRBDs. The correlation factors $Rp$ and the r.m.s.d. values calculated for the different models vary significantly, with values from 0.71 to 0.93 and 4.1 to 8.0 Hz, respectively, indicating that our RDC dataset is a sensitive probe of the relative orientation of the two domains (Fig 3A). For each of the two dsRBD2 positions, two dsRBD1 registers yielded minimal r.m.s.d. and maximal $Rp$ values, and hence were compatible with the RDCs. These registers (#3 and #5, Fig 3A and B, green and magenta) are characterized by antiparallel and parallel arrangements of the two domains, respectively.

To resolve this ambiguity, six inter-domain distance distributions were measured using EPR spectroscopy with SDSL (Jeschke *et al*, 2006; Bordignon, 2012; López *et al*, 2012) and compared with the corresponding distance distributions calculated in the models compatible with the RDCs. As shown by overlaying the experimental and simulated distance distributions, the configurations with the two domains pointing in opposite directions are in good agreement with the EPR data, whereas the configurations featuring parallel domains deviate significantly, especially for the distances Cys63-Cys191, Cys63-Cys197, and Cys65-Cys193 (Fig 3C). Therefore, we conclude that in solution, the relative orientation of dsRBD1 and dsRBD2 is well defined in the dsRBD12-EL86 complex, with the two domains predominantly pointing in opposite directions (anti-parallel orientation). Along with our finding that dsRBD2 has two main binding sites in the dsRBD12-EL86 complex, we come to the conclusion that dsRBD12-EL86 complex formation yields two major species (referred to as complexes "A" and "B" hereafter), in which dsRBD1-dsRBD2 relative orientation is identical.

We then used single-molecule FRET to quantify the relative occurrence of complexes A and B. With single-molecule FRET, the distance-dependent energy transfer between a donor and an acceptor fluorophore is probed, both of which are conjugated to the molecules of interest (Ha & Selvin, 2008). Because single molecules rather than ensemble averages are measured, the method provides an additional possibility to resolve structural heterogeneity and obtain information about the relative occurrence of the underlying configurations. To ensure that the fluorophores do not affect the function of the protein, we first quantified the affinity of labeled dsRBD12 toward EL86 (Fig EV3A and B). Our results ($K_D$ = 210 ± 30 pM) are in agreement with the affinity of unlabeled dsRBD12 for a 21 bp duplex ($K_D$ = 250 pM) previously quantified by isothermal titration calorimetry (Yamashita *et al*, 2011), indicating that the dyes have a negligible influence on TRBP's function. To investigate the conformations of dsRBD12 in complex with EL86, EL86 was labeled at its 3′-end with the FRET donor Cy3B. The FRET acceptor CF660R was site specifically incorporated into dsRBD12 via a Cys residue introduced at position 100 (dsRBD12 M100C/C158S) or using the naturally occurring Cys residue at position 158 (dsRBD12 M100S), respectively. For both dsRBD12 variants, the two different domain arrangements are expected to lead to

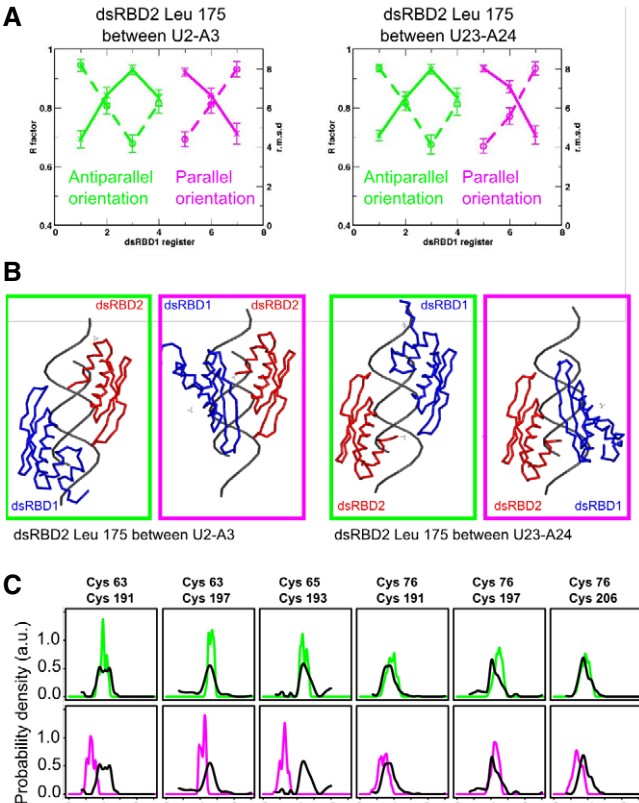

Figure 3.  DsRBD1 positioning relative to dsRBD2 in RNA-bound dsRBD12.

A  RDCs fitting as a function of dsRBD1-dsRBD2 relative orientation. Fourteen models of the dsRBD12-EL86 complex were built (not shown), in which dsRBD2 was fixed in two registers by "anchoring" Leu175 between either Ura2-Ade3 (left) or Ura23-Ade24 (right), and the position of dsRBD1 on EL86 was systematically shifted. R.m.s.d. values (dashed lines) and Pearson's correlation coefficients (solid lines) are depicted for each model. RDC fitting with models where dsRBD1 and dsRBD2 point in opposite or in the same directions are shown in green and magenta, respectively.

B  Each position of dsRBD2 (red) yields two possible binding registers for dsRBD1 (blue), due to the intrinsic degeneracy of RDCs. Solutions in which the two domains are "parallel" or "anti-parallel" are represented in magenta and green frames, respectively.

C  Comparison of EPR (black) and simulated (green or magenta) dsRBD1-dsRBD2 pairwise distances. Inter-domain distances were simulated using either the anti-parallel (upper row, green curves) or the parallel (lower row, blue curves) models of dsRBD12-EL86 shown in (B).

pronounced differences in transfer efficiency because of different inter-dye distances (Fig 4A).

Indeed, as shown in Fig 4B (top), two peaks are observed in the resulting transfer efficiency histograms. For dsRBD12 M100C/C158S, the peak centered at a transfer efficiency of 0.4 corresponds to complex A, and the peak at 0.9 to complex B. In the case of dsRBD12 M100S, the inter-dye distances of the two configurations are more similar, resulting in a more pronounced overlap of the two transfer efficiency peaks at 0.6 (complex B) and 0.9 (complex A), respectively, consistent with the intermolecular distances expected from the structural models (Fig 4B). The relative populations of the two configurations were quantified by peak integration. To obtain accurate positions and shapes of transfer efficiency distributions for

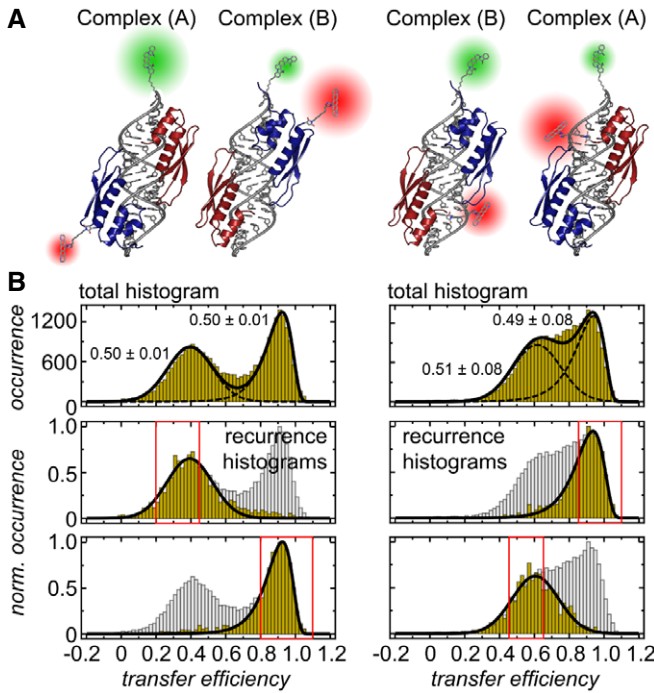

**Figure 4. Quantifying the relative occurrence of the two possible configurations of dsRBD12 on EL86 by single-molecule FRET.**

A Cartoons of CF660R-labeled dsRBD12 C158S (left) and dsRBD12 M100S (right) bound to Cy3B-labeled EL86. The different domain arrangements of dsRBD12 on EL86 are characterized by different inter-dye distances, which were approximated as distances between their attachment points (as M100 is not resolved in the solution structure, the nearest resolved neighbor G98 was used, resulting in an uncertainty of 7 Å, i.e., the contour length of two residues).

B Transfer efficiency histograms of CF660R-labeled dsRBD12_22-235 C158S (left) and dsRBD12_22-235 M100S (right) in complex with Cy3B-labeled EL86. Top: Transfer efficiency histograms exhibit two subpopulations that are equally likely to occur. Errors associated with relative occurrences correspond to the standard deviation. Bottom: Recurrence transfer efficiency histograms were used to extract subpopulation-specific fit parameters. Red boxes highlight the initial transfer efficiency range ΔE. The recurrence interval *T* was set to (0, 10 ms). See Materials and Methods for details.

the fit, we performed recurrence analysis (Hoffmann *et al*, 2011), followed by fitting subpopulation-specific recurrence histograms to empirical fit functions (Fig 4B, bottom). The resulting peak parameters were then used to fit the complete histograms (Figure 4B, top). Based on the peak integrals, we observed very similar populations for complexes (A) and (B), both for dsRBD12 M100C/C158S (0.50 ± 0.01) and dsRBD12 M100S, although the overlap of the peaks resulted in greater uncertainty for the latter (complex (A): 0.49 ± 0.08; complex (B): 0.51 ± 0.08). Very similar results were obtained with a highly asymmetric siRNA (pp-luc, Fig EV3C), and a symmetric siRNA (sod1, Fig EV3D), demonstrating that the orientation of the two dsRBDs and the relative population of the two complex forms is independent of the sequence.

In summary, single-molecule FRET demonstrates (i) that dsRBD12 binding to EL86 results in two major configurations and (ii) that the relative occurrence of these two conformers is very similar. This indicates that the interactions in the two binding modes are

isoenergetic, and the dsRBD12-RNA interactions are thus unlikely to be sequence-specific.

## DsRBD12 binding surface on EL86 resembles a half-cylinder

Using the structural insight gained from NMR and EPR experiments, several ambiguities in intermolecular NOEs assignment could be resolved, providing supplementary distance restraints for structure calculations. The intense intermolecular NOEs from the methyl groups of Ala57 and Ala187, located at equivalent positions in the β1–β2 loops of dsRBD1 and dsRBD2, were assigned to five EL86 resonances: C14 (H1′), C15 (H1′), A35 (H1′), A36 (H1′), and A36 (H2) (Fig 5A). The weaker intermolecular NOEs from Thr30, Thr40 (dsRBD1), Val161, and Val169 (dsRBD2) to EL86 H1′ protons could also be assigned (Fig 5A). Remarkably, Thr40 and Val169, both of which are located at the C-terminus of dsRBD1 and dsRBD2 α1 helices, have identical NOE patterns (C19 H1′ and A40 H1′), further supporting the identical registers of dsRBD1 and dsRBD2 in both complexes. Two sets of distances were derived from these intermolecular NOEs and were used as structural restraints to calculate two structures of the dsRBD12-EL86 complex: In complex A, the dsRBD1 (dsRBD2) β1–β2 loop region was constrained to lie between A35 and A36 (C14 and C15), whereas in complex B, the constraints were inverted, with dsRBD1 (dsRBD2) β1–β2 loop region constrained between C14-C15 (A35–A36). Calculations yielded two well-defined ensembles (Appendix Table S3 and Fig 5B) in which dsRBD1 and dsRBD2 point in opposite directions along the axis of EL86 helix: The α1 helices are located at the extremities of EL86, whereas the β1–β2 loops are located near the center in a head-to-head fashion (Fig 5B). The good precision of the structure ensembles (backbone r.m.s.d. of 0.87 ± 0.18 Å and 0.83 ± 0.15 Å for complexes A and B, respectively, Appendix Table S3) allows for a detailed analysis of the structures.

In the two complexes, dsRBD1 and dsRBD2 bind EL86 via the dsRBD canonical binding surface. The two domains point in opposite directions and the two complexes are very similar, albeit dsRBD1 and dsRBD2 are swapped (Fig 5B). Remarkably, the two domains bind one side of the RNA helix (a half-cylinder) over the whole length of EL86. The remainder of the RNA surface, which is solvent-exposed in our structures, corresponds to the other half-cylinder of the EL86 RNA helix (Fig 5C). The phosphate backbone of EL86 is recognized by the evolutionarily conserved KKxAK motif, corresponding to Lys80, Lys81, Lys84 and Lys210, Lys211, Lys214 in dsRBD1 and dsRBD2, respectively (Fig EV4). The β1–β2 loop of dsRBD2 is well defined and interacts with the minor groove of EL86. In particular, the Ala187 methyl group makes hydrophobic contacts with A35 and A36, and C14 and C15 in complexes A and B, respectively (Fig EV4). In complex A, the carbonyl group of Ala187 is hydrogen bonded to the amino group of G27. In complex A (B), His188 Nδ1 makes a hydrogen bond to the 2′-OH of C15 (A36), whereas the peptide carbonyl group interacts with the 2′-OH of G27 (U6), bridging the two RNA strands across the minor groove, as reported previously for second dsRBD of *X. laevis* RNA-binding protein (Ryter & Schultz, 1998). The β1–β2 loop of dsRBD1 is less well defined, but Ala57 and His58 make essentially the same interactions as Ala187 and His188 (Fig EV4). Residues Ile32, Ser33, Gln36, Glu37, Thr40, and Arg41 located within dsRBD1 helix α1 and residues Val161, Gln165, Glu167, Val169, Gln170 located within

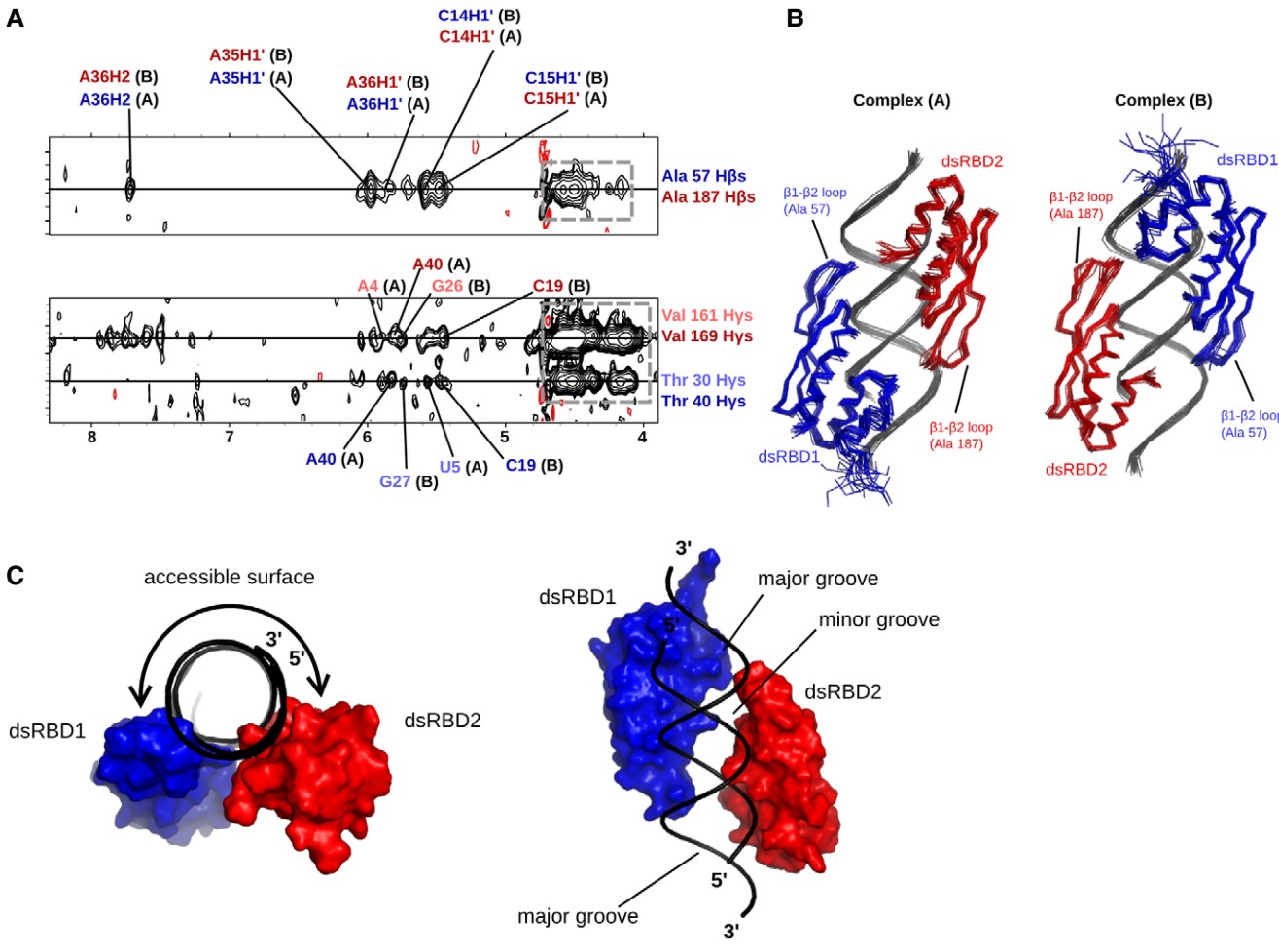

**Figure 5. Three dimensional structures of the dsRBD12-EL86 complex.**

A   Selected regions of the $^{13}$C-edited filtered NOESY spectrum recorded on the dsRBD12-EL86, showing intermolecular NOEs between Ala57 Hβs (dsRBD1)/Ala187 Hβs (dsRBD2) and EL86 H1′ protons. Blue (red) assignment labels correspond to dsRBD1-Ala57 (dsRBD2-Ala187). The particular domain arrangement corresponding to each assignment is designated by A or B, respectively. NOE peaks in the gray dashed box correspond to EL86 H4′/H5′/H5″ protons and carry very limited, if any, sequence-related information.

B   Structure ensembles of complexes A and B calculated using experimental intermolecular distance restraints and residual dipolar couplings. dsRBD1 and dsRBD2 are colored in blue and red, respectively.

C   Global view of the dsRBD12-EL86 complex showing the half-cylinder RNA region left solvent-exposed, and potentially accessible to other proteins. dsRBD1 and dsRBD2 surfaces are represented in blue and red, respectively. EL86 is represented as a black cartoon.

dsRBD2 helix α1 make numerous hydrophobic and polar contacts in the RNA minor groove with ribose moieties (Fig EV4). In addition, two side-chains (Lys29 and Thr30) located in the loop connecting dsRBD1 helices α0 and α1 interact with ribose sugars on each side of the RNA minor groove.

We then used EL86 variants, in which a 2′-O-methoxyethyl group was incorporated at each single position on the guide strand (upper strand in Fig 1A) during chemical synthesis, to knock down a cognate mRNA in HeLa cells. As observed in other studies (Prakash *et al*, 2005; Jackson *et al*, 2006), the knockdown efficiency was affected in a position-dependent manner (Fig EV5). The strongest effects are observed when either position 1, 2, or 14 is modified. Based on our structures, U1 does not interact with dsRBD12, and C14 is contacted by the methyl group of Ala57 (dsRBD1) or Ala187 (dsRBD2), in complexes B and A,

respectively. Furthermore, U2 forms contacts with dsRBD2 Leu175 in complex A (Fig EV5). Consequently, one could speculate that the reduced efficiency might originate from a weakened interaction with TRBP.

We propose that the spatial arrangement of TRBP dsRBDs on EL86 primarily results from the recognition of the structural features of the A-form RNA helix. We note that this particular configuration of the two domains is the only way to position helix α1 at the RNA ends for both domains simultaneously. This is reminiscent of a trend observed in other dsRBD-dsRNA structures, where helix α1 often interacts with stem-loop junctions (widened minor groove), whereas the β1–β2 loop is found in more regular RNA stems (Stefl *et al*, 2010; Wang *et al*, 2011). Finally, it is noteworthy that dsRBD12 covers a continuous RNA region of 19 base pairs resembling a half-cylinder, while the second half of the RNA surface

remains solvent-exposed. It is therefore tempting to speculate that Dicer binds on this surface during pre-miRNA processing.

## DsRBD12 binds shEL86 during Dicer processing under single-turnover conditions

We designed a pre-miRNA (shEL86) by extending EL86 by two additional base pairs and a nine-residue terminal loop (Fig 6A; Bofill-De Ros & Gu, 2016). shEL86 was $^{32}$P-radiolabeled at its 5′-end and used as a substrate for Dicer under single-turnover conditions. Denaturing PAGE analysis reveals two cleavage products that are 22 nt and 21 nt in length, respectively (Fig 6B). These two bands result from the Dicer cleavage of shEL86's 5′ arm at two consecutive positions and display an intensity ratio of 2:1 (Fig 6A). Next, we formed TRBP dsRBD12-shEL86 complexes by incubating 12 nM shEL86 with dsRBD12 at increasing concentrations (5–400 nM), followed by Dicer-mediated cleavage. The dsRBD12-EL86 complex displays a dissociation constant of 0.21 nM (Fig EV3B), and hence, dsRBD12 and shEL86 molecules are mostly associated within complexes under our experimental conditions. At a dsRBD12:shEL86 ratios below 10:1, we do not observe any statistically significant difference in the amount of RNA cleavage products despite a slight cleavage increase in the presence of TRBP up to 1:1 (Fig 6B, lanes 2–6). In contrast, we observe a clear inhibition of product formation at dsRBD12:EL86 ratios greater than 10:1 (Fig 6B, lanes 7–10). We therefore conclude that TRBP dsRBD12 does not interfere with Dicer cleavage of shEL86 unless present in large excess, where two dsRRBD12 molecules are likely to bind simultaneously. These results are in agreement with our structure-based proposal that dsRBD12 and Dicer both bind shEL86 at two distinct and non-overlapping regions during shEL86 cleavage at its terminal loop.

## Modeling of a dsRBD12-EL86-Dicer ternary complex

We then built a structural model of a ternary TRBP-RNA-Dicer complex to understand better how TRBP and Dicer may act during pre-miRNA processing. For this purpose, we docked the X-ray structure of the PAZ-platform-Connector-RNA complex (Tian *et al*, 2014) onto our dsRBD12-EL86 structure by superposing the RNA present in both structures. We then positioned EL86 onto Dicer's RNase III catalytic center in the following way: First, we docked the X-ray structure of the Aa RNase III–RNA complex (Gan *et al*, 2006) onto the PAZ-dsRBD12-EL86 assembly by superimposing the two RNA strands present in the RNase III structure onto the EL86 end distal from the PAZ domain. Second, we overlaid the X-ray structure of Dicer RNase III (Du *et al*, 2008) onto the Aa RNase III structure using well-conserved regions. As a result, we obtained a preliminary model in which TRBP dsRBD12, Dicer's RNase III, PAZ, and platform domains are all docked on the surface of EL86. Remarkably, each domain is found to interact with a distinct EL86 region, and the ternary complex is devoid of steric clashes. Moreover, the model suggests a possible interaction between the C-terminal dsRBD of Dicer with both TRBP dsRBD2 and the RNA minor groove (Fig 7A). In this context, the dsRBD of Dicer does not compete for EL86 with the dsRBDs of TRBP. We then fitted our model into the EM envelope of apo-Dicer (Taylor *et al*, 2013). Here, we could also position the X-ray structures of the first and third domains of RIG-I (Kowalinski *et al* 2011) and of the TRBP-dsRBD3-Dicer-PBD complex (Wilson *et al*, 2015) into the EM density corresponding to the base branch (Lau *et al* 2012). The volume of the Dicer map could accommodate the entire structural model, with the

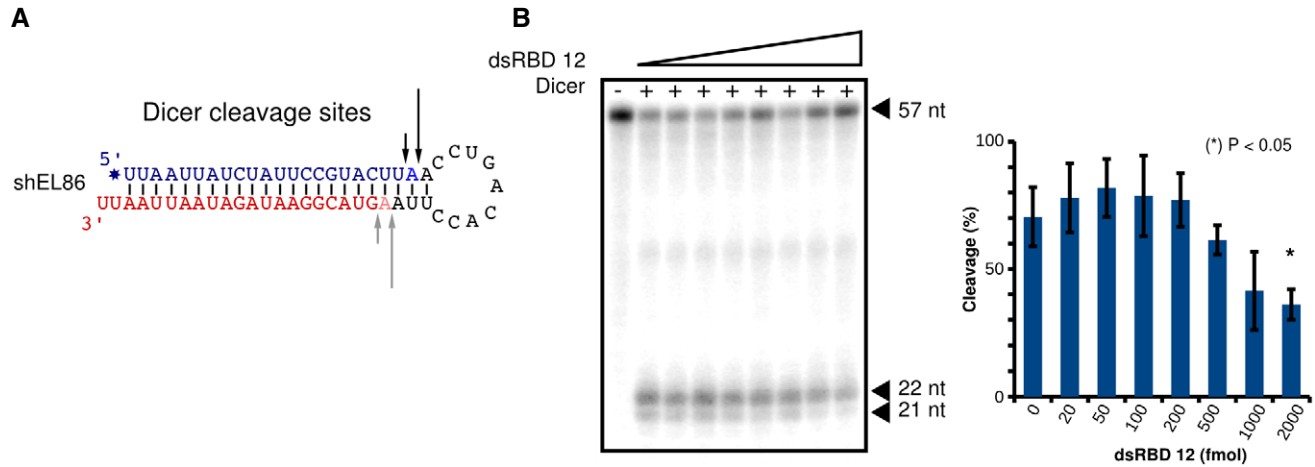

**Figure 6. Dicing assay: shEL86 cleavage by Dicer.**

A Secondary structure of shEL86. The 5′ and 3′ arms are colored blue and red, respectively. The $^{32}$P-radiolabel incorporated at the 5′ end is indicated by a blue asterisk. Dicer's cleavage sites experimentally determined are indicated by black arrows. Gray arrows indicate Dicer's cleavage sites inferred from the cleavage sites positions on the 5′-arm.

B (Left) Representative denaturing polyacrylamide gel showing uncleaved RNA substrate (57 nucleotides) and two cleavage products of 21 and 22 nucleotides. Lane 1: negative control in the absence of Dicer; lanes 2–9: Dicer cleavage in the presence of increasing concentration of dsRBD12 (0, 20, 50, 100, 500, 200, 1,000, 2,000 fmol). (Right) Quantification of shEL86 cleavage for increasing dsRBD12 concentrations, calculated as 100× RNA-product: total-RNA. Each datapoint represents the average ± SD of three experimental replicates. The cleavage yields in the absence and in the presence of dsRBD12 were compared with a Student's *t*-test. The asterisk denotes a *P*-value of < 0.05.

exception of TRBP dsRBD12, as expected. In particular, the relative orientation of the PAZ-RNase III domains, obtained indirectly by overlaying RNA molecules from different structures, matches the EM density well (Fig 7B). We arbitrarily chose to orient our dsRBD12-EL86 structure on Dicer's RNase III domains such that dsRBD1 and dsRBD2 are located near the PAZ and the helicase domains, respectively. In our model, the distances between the N-terminus of dsRBD3 and the C-terminus of TRBP dsRBD1 and dsRBD2 are 75 and 40 Å, respectively. DsRBD2 and dsRBD3 are connected by a linker of 30 amino acids, which can in theory span up to 108 Å in an extended conformation. Our model is therefore compatible with the length of TRBP dsRBD2-3 inter-domain linker. It does not permit, however, to predict the relative orientation of two N-terminal dsRBDs of TRBP during pre-miRNA processing.

## Discussion

### Role of TRBP in mi/siRNA asymmetry

Guide strand selection is a crucial process, since it determines which mRNAs are to be translationally repressed in the cytoplasm. Likewise, failure to accurately predict the guide strand when designing a siRNA will result in "off-target" effects (Bofill-De Ros & Gu, 2016). The molecular mechanism governing strand selection is not well understood, albeit it is generally agreed that the fate of the two si/miRNA strands is determined after mi/siRNA duplex loading into Argonaute. The stability of the mi/siRNA duplex termini has been proposed to be one of the determinants of strand selection and is based on the observation that the RNA strand with the less stable 5′ end is usually selected as the guide (Khvorova *et al*, 2003; Schwarz *et al*, 2003). The heterodimers R2D2/Dcr-2 and Loqs/Dcr-2 in flies, and Dicer/TRBP in humans, are able to bind asymmetric siRNA duplexes with a well-defined orientation (Tomari *et al*, 2004; Noland *et al*, 2011; Tants *et al*, 2017). It has also been shown that human Dicer is dispensable for asymmetric RISC assembly (Betancur & Tomari, 2012). Consequently, the proteins TRBP (Gredell *et al*, 2010) and its Drosophila homolog R2D2 (Tomari *et al*, 2004) were proposed to transduce si/miRNA thermodynamic asymmetry into strand selection. The protein Ago has also been shown to play a role in guide strand selection, and structural studies have unveiled a direct readout of the 5′ nucleobase by Ago MID domain (Frank *et al*, 2010; Suzuki *et al*, 2015).

Here, we studied the interaction of TRBP's N-terminal dsRBD12 (Yamashita *et al*, 2011; Benoit *et al*, 2013) with a highly asymmetric siRNA (EL86) exhibiting a strand selection bias of several orders of magnitude (Stalder *et al*, 2013). Our results show that these two molecules form two distinct complexes in solution that are equally populated. Essentially, the two complexes differ by a 180° re-orientation of EL86 with respect to the dsRBD12, or equivalently, by a swapping of dsRBD1 and dsRBD2 binding sites. Very similar results are obtained for the asymmetric siRNA pp-luc, and the symmetric siRNA sod1 (Fig EV3C and D). We therefore propose that TRBP's unbiased binding behavior is not specific to EL86. Rather, it appears to be a general property of the protein. Within our experimental setting, siRNA asymmetry does not influence TRBP binding, which

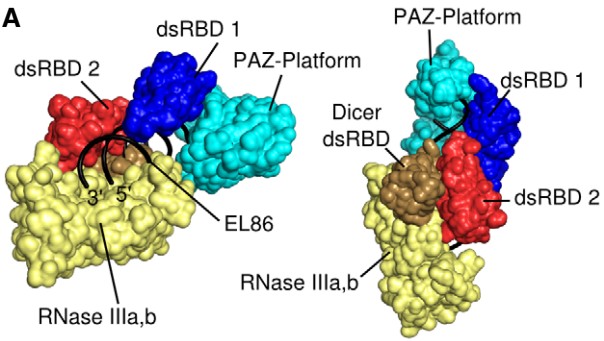

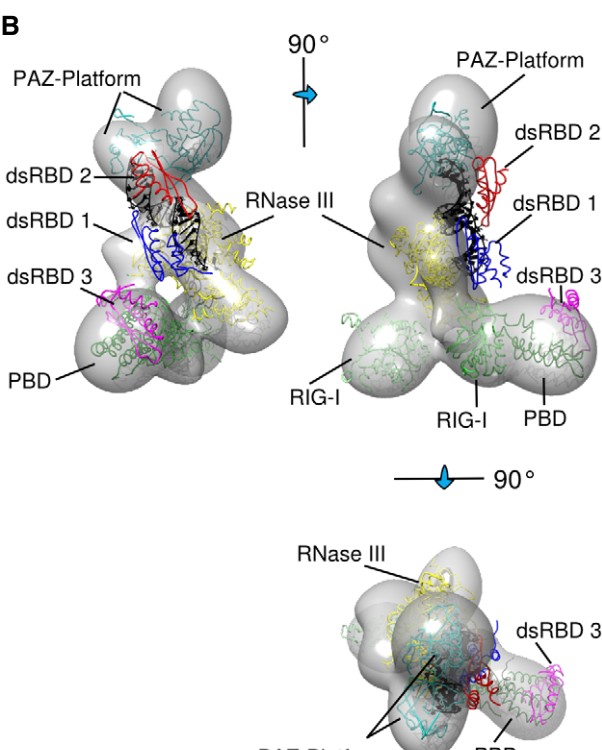

**Figure 7.  Three dimensional modeling of a putative ternary complex, showing shEL86 cleavage product binding simultaneously to Dicer and TRBP dsRBD12.**

A  (Left) Spatial arrangement of Dicer's RNase III (yellow, PDB id 3C4B; Du *et al*, 2008) and PAZ-Platform (cyan, PDB id 4NHA; Tian *et al*, 2014) domains resulting from their docking onto EL86 (black). Each domain interacts with a distinct RNA region without any steric clash. (Right) Side-view showing the proximity of Dicer's dsRBD (gold) with TRBP dsRBD2 (red). dsRBD1 is shown in blue.

B  The distance between TRBP dsRBD2 and dsRBD3 was estimated by docking within the EM envelope of apo-Dicer (EMD-5601; Taylor *et al*, 2013) our 3D model for the dsRBD12-EL86-PAZ-Platform-RNase III, RIG-I domains 1,3 (PDB id 4A36; Kowalinski *et al*, 2011) and TRBP-dsRBD3-Dicer-PBD (PDB id 4WYQ; Wilson *et al*, 2015). Orthogonal views are shown.

suggests that TRBP *alone* cannot "sense" siRNA or miRNA asymmetry. It cannot be excluded, however, that TRBP participates to asymmetric si/miRNA loading into Ago. It has been reported in a very recent study that in flies, the heterodimer Loqs-PD-Dcr2 is able to

discriminate the two ends of an asymmetric RNA. This ability relies on a moderate intrinsic binding preference of Loqs-PD for the most stable end of a 21-bp siRNA (Tants *et al*, 2017). This apparent difference in the mode of binding of these functionally homologous proteins with regard to RNA asymmetry sensing is likely to be caused by a sequence divergence between the two proteins, in particular within the residues binding RNA and in the length of the inter-domain linker, which is shorter in Loqs-PD. Additionally, the longer siRNA substrate used in the work by Tants *et al* (two base pairs longer) might also influence the mode of binding.

### TRBP's influence on pre-miRNA processing

The processing of pre-miRNA by Dicer, which consists in the excision of the pre-miRNA terminal loop, is regulated, at least partly, by TRBP. *In vitro*, TRBP has been shown to accelerate or inhibit the pre-miRNA cleavage rate (Lee & Doudna, 2012), and to maintain pre-miRNA efficient processing in a RNA-crowded environments (Fareh *et al*, 2016). Furthermore, for several pre-miRNAs, TRBP shifts the position of the Dicer-cleavage site, resulting in miRNAs one nucleotide longer (Fukunaga *et al*, 2012; Lee & Doudna, 2012; Wilson *et al*, 2015). This change of the 5′ nucleotide of the miRNA's 3′ arm can lead in some particular cases to an inversion of the guide/passenger strands (Kim *et al*, 2014; Wilson *et al*, 2015). The molecular mechanism whereby TRBP produces its effects is not well understood yet. However, the simplest hypothesis is that TRBP interacts with the pre-miRNA and/or Dicer, during pre-miRNA processing. The siRNA used in this study (EL86) has 19 base pairs and 2-nucleotide 3′ overhangs at both ends. It is therefore very similar, with regard to its size and secondary structure, to a miRNA product resulting from the processing of a pre-miRNA. Interestingly, our structure of TRBP dsRBD12 in complex with EL86 shows that dsRBD12 interacts with one face of the RNA, covering a surface resembling a half-cylinder (Fig 5). Our dicing assay performed on shEL86, an RNA hairpin derived from EL86 and mimicking a pre-miRNA, carried out in the presence or in the absence of TRBP, shows that TRBP does not compete with Dicer for binding shEL86 (Fig 7). We propose therefore that during pre-miRNA processing, Dicer and TRBP dsRBD12 bind the pre-miRNA simultaneously, on two distinct, non-overlapping, surfaces. Using the structures of various Dicer's fragments in complex with RNA along with our structure of TRBP-dsRBD12 in complex with EL86, we built a structural model of a ternary Dicer-EL86-dsRBD12 complex, which mimics a reaction intermediate resulting from pre-miRNA dicing (Fig 7A). Furthermore, our model suggests that TRBP-dsRBD12 can bind the miRNA region of a pre-miRNA, without causing steric clashes with Dicer. This model provides therefore a physically realistic description of a putative Dicer-miRNA-dsRBD12 complex, which suggests that TRBP's effect on the size of the miRNA product could result from dsRBD12 capacity to affect the structure of the pre-miRNA's stem (e.g., by stabilizing looped-out nucleotides) (Fukunaga *et al*, 2012; Lee & Doudna, 2012; Kim *et al*, 2014; Wilson *et al*, 2015). According to our model, the formation of a stable Dicer-pre-miRNA-dsRBD12 complex competent for pre-miRNA processing necessitates a pre-miRNA with a dsRNA length of 19 base pairs. This is in agreement with a previous report demonstrating that TRBP acts as a gate-keeper, preventing Dicer to engage in stable interactions with non-cognate substrates (Fareh *et al*, 2016).

## Materials and Methods

### Protein expression and purification

Amino acid numbering of the various TRBP fragments used in this study refers to the wild-type human TRBP (UniProtKB Q15633). In brief, DNA fragments encoding TRBP dsRBD1 (16–108), dsRBD2 (150–227), or dsRBD12 (16–227) were inserted in pet28a vectors, modified to include a tobacco etch virus (TEV) protease cleavage site immediately upstream of the multi-cloning site. Proteins were overexpressed in *Escherichia coli* BL21(DE3) Codon plus (RIL) cells using standard techniques. Hexa-histidine tags were removed by TEV digestion. A detailed purification protocol can be found in Methods section within the Appendix.

### NMR spectroscopy and structure calculations

All the NMR experiments were recorded at 313 K. Data were processed using Topspin 3.1 (Bruker) and analyzed with Sparky (T. D. Goddard and D. G. Kneller, SPARKY 3, University of California, San Francisco). Spectra analysis and structure calculation were carried out using standard procedures described elsewhere (Dominguez *et al*, 2011). Further details are provided in the Methods section within the Appendix.

### EPR spectroscopy and DEER analysis

The samples for DEER measurements were prepared as solutions of *ca.* 100 μM of protein–RNA complex in 1/1 $D_2O$/D-glycerol (v/v) mixture. For each sample, about 30 μl of such mixture was placed into a quartz tube of 3 mm outer diameter and frozen by immersion into liquid nitrogen and stored until measurement. Further details concerning instrumentation, data acquisition and data analysis are to be found in Methods section within the Appendix.

### RNA preparation

EL86 oligonucleotides used in single-molecule FRET experiments were purchased RP-HPLC-purified from Integrated DNA Technologies BVBA (Leuven, Belgium), where EL86up (5′-UUA AUU AUC UAU UCC GUA CUU-3′) was functionalized with a biotin at its 3′-end, while a primary amino modifier was incorporated at the 3′-end of EL86down (5′-GUA CGG AAU AGA UAA UUA AUU-3′). Oligonucleotides used in NMR and EPR experiments were chemically synthetized on an Äkta Oligopilot plus OP100 (GE Healthcare). Further information is provided in the Methods section within the Appendix.

### Single-molecule spectroscopy

Single-molecule FRET experiments were conducted at 295 K on freely diffusing molecules with a custom-built confocal microscope equipped with a UplanApo 60×/1.20 W objective (Olympus) and a 100-μm pinhole. Fluorophores were excited alternatingly using pulsed interleaved excitation (Müller *et al*, 2005) with light of a supercontinuum fiber laser (SC-450-4, Fianium Ltd., Southampton, UK) filtered by a HC543.5/2 band pass for donor excitation and a diode laser emitting at 640 nm (LDH-D-C-640, Picoquant GmbH, Berlin, Germany) for acceptor excitation. Both lasers were operated

at a repetition rate of 20 MHz, and the intensities were adjusted to 50 μW at the back aperture of the objective. Fluorescence emitted by the sample was collected by the objective and separated according to polarization using a polarizing beam splitter, followed by separation according to wavelength with two dichroic beam splitters (635DCXR, Chroma Technology GmbH, Olching, Germany). Donor detection channels were equipped with ET585/65 m band-pass filters (Chroma Technology GmbH, Olching, Germany) and τ-SPAD avalanche photodiodes (Picoquant, Berlin, Germany). Acceptor detection channels were equipped with LP 647 RU long band-pass filters (Semrock, Inc. Rochester, NY) and SPCM-AQRH-14 avalanche photodiodes (PerkinElmer AG, Schwerzenbach, Switzerland). Photon arrival times were recorded with a Hydraharp 400 time-correlated single-photon counting system (PicoQuant GmbH, Berlin, Germany) at a time resolution of 16 ps. All measurements were conducted in PEGylated sample chambers (Microsurfaces, Inc., Eaglewood, NJ, USA). To study the orientation of dsRBDs on dsRNA, sample solution containing 10 pM Cy3b-labeled RNA (EL86down, sod1down, or pplucdown) and 333 nM unlabeled complement (EL86up, sod1up, or pplucup), 66 pM CF660R-labeled dsRBD12, 15 nM BSA, 0.001% (w/v) Tween-20, dissolved in 20 mM Tris–HCl, 25 mM KCl, pH 7.4 were used. Single-molecule data were recorded for 8–14 h to ensure sufficient statistics for recurrence analysis. To quantify the affinity of dsRBD12 toward dsRNA, sample solutions containing unlabeled EL86, 25 pM Cy3b-CF660R-labeled dsRBD12, 15 nM BSA, 0.001% (w/v) Tween-20, dissolved in 20 mM Tris–HCl, 125 mM KCl, pH 7.4 were measured for 1 h.

## Single-molecule data analysis

Photon bursts emitted from fluorescently labeled molecules diffusing through the confocal volume were identified as contiguous intervals of emission with inter-photon times of less than 150 μs and a minimum number of photons of 30. Subsequently, bursts were corrected for differences in chromophore quantum yields, differences in detection efficiency of the detectors, spectral crosstalk, direct acceptor excitation, and background signal (Schuler, 2007). The stoichiometry ratio, $S$, of a burst was calculated according to

$$S = \frac{n_{\text{tot,Dex}}}{n_{\text{tot,Dex}} + n_{\text{tot,Aex}}}$$

where $n_{\text{tot,Dex}}$ and $n_{\text{tot,Aex}}$ denote the corrected total number of photons emitted after donor or acceptor excitation, respectively (Müller *et al*, 2005). Bursts with $0.3 < S < 0.65$ were used to calculate the transfer efficiency $E$ as

$$E = \frac{n_{\text{A}}}{n_{\text{A}} + n_{\text{D}}}$$

where $n_{\text{D}}$ and $n_{\text{A}}$ are the corrected donor and acceptor photon counts upon donor excitation within a burst. Transfer efficiencies were binned in histograms.

To obtain subpopulation-specific transfer efficiency histograms for probing the orientation of dsRBDs on dsRNA, we used recurrence analysis of single particles (RASP) (Hoffmann *et al*, 2011). RASP relies on the observation that at sample concentrations in the low picomolar range and for timescales up to several tens of milliseconds, the probability that two consecutive fluorescent bursts

originate from the same molecule is higher than that they stem from different molecules. Provided that conformational dynamics occur on a timescale much slower than the diffusion time, two consecutive bursts are therefore likely to yield the same transfer efficiency. Based on this recurrence behavior, subpopulations were isolated from each measurement using an initial transfer efficiency range $\Delta E$ (indicated as red boxes in Figs 4 and EV3C and D) and a recurrence interval between 0 and 10 ms. The resulting recurrence histograms were fitted to Gaussian or lognormal peak functions to determine their positions and shapes. These fit functions were then used to describe the complete histograms (8–14 h measurements), where only the peak amplitudes were allowed to vary. The relative occurrences of subpopulations were calculated from the resulting peak areas. The corresponding standard deviations were estimated by splitting the recorded data into segments of 1 h, followed by determination of subpopulation-specific relative occurrences through constrained fitting using the peak parameters obtained by RASP.

To quantify the affinity of dsRBD12 toward dsRNA, a binding titration was performed. Transfer efficiency histograms of FRET-labeled TRBP (~25 pM) recorded at 0 and 100 nM dsRNA were described with a single Gaussian (unbound) or lognormal peak function (bound), respectively, and the resulting positions and shapes of the fit functions were fixed for a global fit of all transfer efficiency histograms in terms of a two-state model. As a result, the peak amplitudes of the bound and unbound states are the only adjustable parameters in the fit. The fractional occupancy of unbound and bound states was quantified from the relative peak areas, followed by fitting the data to the binding isotherm

$$\theta = \frac{c_{\text{TRBP}} + c_{\text{dsRNA}} + K_{\text{D}} - \sqrt{(c_{\text{TRBP}} + c_{\text{dsRNA}} + K_{\text{D}})^2 - 4\,c_{\text{TRBP}}\,c_{\text{dsRNA}}}}{2\,c_{\text{TRBP}}},$$

where $\theta$ is the fraction of TRBP bound to dsRNA, $c_{\text{TRBP}}$ and $c_{\text{dsRNA}}$ are the total concentrations of TRBP and dsRNA, respectively, and $K_{\text{D}}$ is the dissociation constant.

## Dicing assays

Single turnover shEL86 cleavage by Dicer was carried out as previously reported (Ma *et al*, 2012). Recombinant human Dicer variant 1 (NCBI Accession No. NM_177438) was purchased from OriGene Technologies, Inc. 9620 Medical Center Drive, Suite 200 Rockville, MD 20852, Catalog No. TP319214. The nucleotide sequence of shEL86 is 5′-UUAAUUAUCUAUUCCGUACUUAACCUGACACCUUAA GUACGGAAUAGAUAAUUAAUU-3′. shEL86 was 5′-end labeled using [γ$^{32}$P] ATP (Hartmann Analytic GmbH) and T4 polynucleotide kinase (New England Biolabs Inc.). Dicing reaction was performed in 5 μl of reaction mix consisting of 25 mM Tris (pH 7.0), 25 mM NaCl, 2 mM DTT, 1.5 mM MgCl$_2$, 1% glycerol, 12 nM shEL86, 120 nM Dicer, 0–400 nM dsRBD12. When present, dsRBD12 was incubated for 15 min with shEL86 prior to Dicer addition. After incubating 30 min at 37°C, the reaction was stopped with 5 μl of loading buffer (10% glycerol, 20 mM TBE, 6 M urea, 0.1% bromophenol blue) followed by heating at 85°C for 10 min. Substrate and cleavage products were resolved by electrophoresis (16% polyacrylamide, urea 7 M). Gel was dried, and RNA substrate and cleavage products were quantified using a Typhoon Trio (GE Healthcare).

## Data availability

The structure coordinates of complex A and complex B have been deposited in the Protein Data Bank (http://www.rcsb.org) and assigned the identifiers 5N8M and 5N8L, respectively.

**Expanded View** for this article is available online.

## Acknowledgements

This work has been supported by the CTI grant 11329.1 PFLS-LS to FHTA and NMK as well as a Sinergia grant from the Swiss National Science Foundation CRSII5_170976 to B.S., F.H.-T.A and G.J. F.A. also acknowledges support from the NCCR RNA and Disease from the Swiss National Science Foundation. We thank Prof. Jonathan Hall (ETH Zürich) for the provision of pre-miRNA samples, Daniel Nettels (University of Zürich) for help with single-molecule FRET analysis and instrumentation. B.S. was supported by the Swiss National Science Foundation.

## Author contributions

GM, CM, and FH-TA designed NMR experiments; GM and CM conducted NMR experiments. MY, GM, and GJ designed EPR experiments; MY conducted EPR experiments. SLBK and BS designed smFRET experiments; SLBK conducted smFRET experiments. GM, CM, and AH prepared protein samples. JM, JH, and NM-K designed and synthetized RNA oligonucleotides. GM designed and performed dicing assays. FA and JW designed and conducted RNA silencing assays. All authors analyzed data. GM, CM, SLBK, BS, MY, JH, and FH-TA wrote the manuscript. ALM synthesized the shEL86 RNA.

## Conflict of interest

The authors declare that they have no conflict of interest.

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
