## [Review Process File · The EMBO Journal]

Structural basis of siRNA recognition by TRBP double-stranded RNA binding domains

Gregoire Masliah, Christophe Maris, Sebastian L.B. König, Maxim Yulikov, Florian Aeschmann, Anna L. Malinowska, Julie Mabile, Jan Weiler, Andrea Holla, Juerg Hunziker, Nicole Meisner-Kober, Benjamin Schuler, Gunnar Jeschke and Frederic H.-T. Allain.

Review timeline:

Submission date:	6 th April 2017
Editorial Decision:	25 th May 2017
Revision received:	11 th December 2017
Editorial Decision:	5 th January 2018
Revision received:	10 th January 2018
Accepted:	12 th January 2018

Editor: Anne Nielsen.

Transaction Report:

1st Editorial Decision

25th May 2017

Thank you for submitting your manuscript for consideration by the EMBO Journal. It has now been seen by three referees whose comments are shown below.

As you will see from the reports, the referees highlight the technical quality of your work and express interest in the findings reported; however, they differ in their views on the overall advance provided by the study. Referee #1 is largely positive but raises a number of clarification points for the analysis and asks for a better integration with the current literature. Ref #2 acknowledges the quality of the work but finds that additional experimentation is needed to establish the generality and functional relevance of the observations made with the current model siRNA. These concerns are shared by ref #3 who goes one step further and finds that the study, while overall technically sound, would be better suited for a more structure-focused journal.

Based on these contrasting views from the referees we have now rediscussed the manuscript in the editorial team and the outcome is that I would invite you to submit a revised version, given the overall positive view from refs #1 and #2 that this is an important and long-standing question in the field. However, in light of the critical comments from ref #2 and #3 I have to ask you to include new experimental data to address all points raised by the referees. In my view, the strategy that ref #2 outlines for adding functional support to the structure data is reasonable and constructive and it should also go some way to address the hesitations expressed by ref #3. You also need to discuss the implications for TRBP in RISC loading/function more critically in the context of the literature, as requested by both refs #1 and #3. I should add that it is EMBO Journal policy to allow only a single round of revision, and acceptance of your manuscript will therefore depend on the completeness of your responses in this revised version.

When preparing your letter of response to the referees' comments, please bear in mind that this will form part of the Review Process File, and will therefore be available online to the community. For more details on our Transparent Editorial Process, please visit our website:
http://emboj.embopress.org/about#Transparent_Process

Thank you for the opportunity to consider your work for publication. I look forward to your revision.

 REFEREE REPORTS

Referee #1:

This mechanism of guide strand selection is a long-standing question in the small RNA field. After the discovery of asymmetry rules, the *Drosophila* RLC, containing Dicer and R2D2, emerged as the most likely asymmetry sensor. Shortly after, several groups suggested that the corresponding Dicer-TRBP complex may serve a similar function in human cells. Since that time evidence supporting and opposing this model has emerged, but a clear and definitive view remains to be established. In this manuscript, Masliah and co-workers provide a beautiful biophysical dissection of the dsRNA-binding properties of human TRBP, showing in convincing detail how TRBP recognizes RNA duplexes, and providing compelling evidence that TRBP does not respond to siRNA asymmetry. Considering the length of time this question has lingered in the field and the substantial technical challenges surrounding addressing the issue with certainty, I feel this work is a very important contribution. As I have limited expertise in spectroscopy methodology, I can mainly offer suggestions to modify the manuscript for improvement:

1) I do not understand the purpose of the chemical modification data presented on pages 14-15. I assume these data were included to strengthen the paper, but as it stands, I believe they do the opposite. A major finding of the study is that dsRBDs 1 and 2 bind equally well to both ends of an asymmetric siRNA duplex. This result leads to a very clear and important conclusion: TRBP is not a sensor of siRNA asymmetry. The chemical modification data clearly suggest asymmetry is at play in the knock-down experiments as "Chemical modification of the passenger strand residues yielded milder effects and a different profile from that obtained for the guide strand". Thus, I would naturally conclude that the chemical modification effects do NOT arise via perturbed interactions with TRBP. I therefore believe the statement, "we observe a clear correlation between how modifications in a number of residues of EL86 reduce translation repression and the presence of intermolecular contacts for these residues in our TRBP dsRBD12 structures" is not in keeping with the other findings in the study. If the authors want to make this claim they should observe interactions between TRBP12 and the modified siRNAs, as they did using EL86. However, it seems more likely to me that these modifications are impacting another step in the silencing process, such as interactions with Ago2, which have been shown to be sensitive to bulky 2' modifications in 5' end of the siRNA seed region (Schirle 2016, PMID: 27380263; Prakash 2005, PMID: 15974578).

2) Figure 6: Although *Giardia* Dicer is the highest resolution Dicer structure currently available, this protein lacks the C-terminal dsRBD and helicase domains present in human Dicer. Therefore, the paucity of steric clashes in the presented TRBP-Dicer model is perhaps not so remarkable. Several moderate resolution EM structures of human Dicer (Lau 2012 PMID: 22426548; Taylor 2014, PMID: 23624860), as well as a structure of TRBP-dsRBD3 bound to a domain from the human Dicer helicase (Wilson 2014, PMID: 25557550), are available. Integrating the new TRBP12 model with these structures would be far more interesting and informative.

3) Discussion section: regarding guide strand selection the authors write of, "two main directions have been proposed to date..." with 1) thermodynamic asymmetry being detected before loading into Ago; and, 2) 5' nucleotide identity being used as a determinant by Ago upon loading. I do not believe this rigid dichotomy accurately reflects current thinking in the field. Noland 2013 (PMID: 23531496) suggest that Ago2 and Dicer-TRBP both contribute to guide selection, and Suzuki, 2015 (PMID: 26098316) suggest that Argonaute senses both thermodynamic asymmetry and 5' nucleotide identity. Therefore, my sense is that the general thought in the field is that there may be multiple steps that contribute to guide strand selection. The difficult is determining the extent to which each putative step contributes, and the mechanisms underlying the different steps. I suggest rewording this part of the Discussion to provide a more holistic representation of the field.

4) Top of Page 4, "While the miRNA duplex is loaded into Ago, only the guide strand is retained..." consider including references to Leuschner PJ, 2006 (PMID: 16439995), and Martinez, 2006 (PMID: 12230974).

5) Page 4, "While in flies the heterodimer Dicer-2/R2D2 was shown to be responsible for asymmetric RISC loading (Tomari, 2007)..." this statement should be qualified with a reference to Nishida, 2013 (PMID: 23375501), which showed that "R2D2 is not absolutely required for siRNA strand selection in vivo."

6) Page 4, Suzuki, 2015 (PMID: 26098316) should be cited when discussing previous work towards understanding guide strand selection in human RISC in the Introduction and Discussion sections.

7) Page 13, "the spatial arrangement of TRBP dsRBDs on EL86 primarily results from the recognition of the structural features of the A-form RNA helix..." How would binding be impacted by mismatches and bulges, commonly found in pre-miRNAs?

8) Page 4/5: It is implied that human TRBP is used, but never actually stated until the Method section on Page 18.

9) Page 5: "we built structural models showing that TRBP has the potential to bind pre-miRNA prior [sic] Dicer cleavage". Neither human Dicer, nor a canonical pre-miRNA are used for the structural models, making this statement slightly inaccurate.

10) The supplemental material lacks a methods section for the cellular assays, particularly with respect to the RT-qPCR (method, internal standard, calculations).

11) Fareh et al., 2016 is cited in the Discussion but does not appear in the References.

Referee #2:

Masliah et al.

Structural basis of siRNA recognition by TRBP double-stranded RNA binding domains

In this manuscript, the authors have used a number of biophysical/biochemical approaches to characterize the interaction between a model siRNA duplex and the first two double stranded RNA binding domains of TRBP. They first investigated interactions of each individual domain with the siRNA duplex using NMR spectroscopy and NOESY experiments. They further solved the solution structure of each domain in the RNA-bound state using NMR. Here, they find that both domains are highly similar except of a short alpha helix that is only present on the N-terminus of dsRBD1 (helix alpha0). Again using NOESY and NMR spectroscopy, the authors characterize RNA interactions in a construct containing the first two dsRBDs as well as the natural linker between these two domains (which does not contact the RNA). They find that the dsRBD2 can bind the siRNA in two different 'registers' (either U2-A3 or U23-A24). Furthermore, dsRBD1 and 2 appear to adopt an antiparallel orientation on the siRNA and the binding is sequence-independent but may rather recognize the shape of the duplex. Masliah et al. used all biophysical data that they have gathered and calculated the structure. They obtain two complexes (A and B), which appear to be in inverted orientations. Finally, they model a complex composed of the siRNA, the two-domain construct and Dicer (based on the crystal structure from *Giardia* Dicer). To validate their structure in vivo, they modified the siRNA at different positions and measured silencing activities. The resulting silencing data suggests that some of the observed interactions are indeed relevant for siRNA function.

This is a solid and competently performed structural analysis of TRBP-siRNA recognition. The authors use a plethora of state-of-the-art biophysical strategies to measure molecular interactions, distances etc. resulting in highly relevant structural information (most importantly, the clear data towards sensing asymmetry in the duplex). However, the study lacks clearer in vivo validation experiments. Specific points are listed below.

1. The authors use a highly asymmetric siRNA and do not observe any effects on the orientation of

the two dsRBDs suggesting that TRBP is not involved in strand selection. This is clearly and important information, which will be highly valuable to the field. Nevertheless, it would be nice to see this analyzed in a more systematic way. Is it possible to study a symmetric and a highly asymmetric siRNA in parallel? This would clearly experimentally strengthen the suggested model.

2. $\alpha 0$ is suggested to help restricting the registers that can be used. This is suggested by the structure but not experimentally validated. Deletion of this helix could result in dramatically changed distance measurements, which would again experimentally validate the results.
3. The authors suggest that TRBP recognizes the 'shape' of the duplex siRNA. What does that exactly mean? Is the shape of a siRNA at the ends different than in the central region? Is there any data that could be used as reference for such a statement?
4. The authors observe two different orientations in their structural model (complex A and B). Is it possible to include the structural information about the third dsRBD in complex with the helicase domain of Dicer (Wilson et al.)? Maybe new constraints appear that would favor one or the other complex? This should be analyzed and added if possible.
5. Figure 6A is not referenced in the text.
6. The validation data is not very strong. The authors even state in the results section that "...efficiency may stem from different potential causes...". Therefore, as presented, it does not contribute much. It would be clearer if for example Ago loading would be analyzed. In this case, at least target binding properties of the modified siRNA strands could be ignored.

Referee #3:

In this manuscript, Masliah et al. report the structure of the N-terminal two dsRBDs of TRBP (but lacking the 3rd dsRBD at the C-terminal) in complex with an siRNA duplex. The combination of NMR, EPR and single-molecule analysis nicely and convincingly shows the detailed tertiary structure of the dsRBD1,2-siRNA complex. However, the functional/biological insights that can be gained from the structure is rather limited and the discussions are often highly speculative. In my opinion, this manuscript is suitable for a more specific, structure-focused journal

Major points:

1. Previous studies have demonstrated that the main function of TRBP is to improve the accuracy of Dicer cleavage site on a subset of pre-miRNAs. Unfortunately, however, the current structure does not provide direct mechanistic explanation for this function of TRBP. Rather, all the related discussions are highly speculative based on the dsRBD1,2 structure (lacking the C-terminal dsRBD) with an siRNA duplex (not a pre-miRNA hairpin) and the docking model with Giardia Dicer (much smaller than human Dicer). Thus, the functional insights from the current structure are quite limited. Fig. 6 is too speculative and should at least be sent to Supplementary Information.
2. It is interesting that the two dsRBDs binds equally to the two ends of the siRNA duplex, in contrast to earlier studies reporting that TRBP can sense the thermodynamic asymmetry of small RNA duplexes. However, by using TRBP knockout cell lines, it has been demonstrated that the effect of TRBP on the guide strand selection is mostly indirect, via the above-mentioned shift of the Dicer cleavage site on a subset of pre-miRNAs (Kim et al., Cell Reports 2014). Thus, the functional importance of the binding mode between TRBP and an siRNA duplex is obscure (unlike the well-established role of Dicer-2/R2D2 binding to an siRNA duplex in Drosophila). Again, this limits the biological insights that can be gained from the current structure.

Minor Points:

1. The authors only used a single siRNA duplex EL86. It will be safe to use at least one more highly asymmetrical siRNA duplex with completely different sequence before making a conclusion that dsRBD1,2 binds to siRNAs symmetrically. Also, direct comparison between NMR and biochemical analysis (i.e., crosslinking) using the same siRNA duplexes will be helpful to fill the gap between the current study and previous reports.
2. Although single-molecule data are beautiful, the possibility that the introduction of fluorescence

dyes affects the proper TRBP function cannot be excluded. Functional comparison between wild-type TRBP and dye-labeled TRBP should be conducted. The possibility that the 3'-end Cy3 modification of the siRNA affects TRBP binding should also be tested.

3. The data in Fig. S4 is problematic; there is no repetition and statistical analysis. In addition, the authors should note that guide strand modification may inhibit the siRNA function at step(s) other than TRBP binding, including Ago loading or RISC maturation.

4. The NMR data are somewhat difficult for non-specialists to interpret. More comprehensive explanation is recommended (e.g., in Fig. 3, how did the authors build 14 different models? How did they narrow down to two possible candidates indicated as green and magenta in Fig. 3B?).

1st Revision - authors' response

11th December 2017

Author Point-by-Point response.

Referee #1:

This mechanism of guide strand selection is a long-standing question in the small RNA field. After the discovery of asymmetry rules, the *Drosophila* RLC, containing Dicer and R2D2, emerged as the most likely asymmetry sensor. Shortly after, several groups suggested that the corresponding Dicer-TRBP complex may serve a similar function in human cells. Since that time evidence supporting and opposing this model has emerged, but a clear and definitive view remains to be established. In this manuscript, Masliah and co-workers provide a beautiful biophysical dissection of the dsRNA-binding properties of human TRBP, showing in convincing detail how TRBP recognizes RNA duplexes, and providing compelling evidence that TRBP does not respond to siRNA asymmetry. Considering the length of time this question has lingered in the field and the substantial technical challenges surrounding addressing the issue with certainty, I feel this work is a very important contribution. As I have limited expertise in spectroscopy methodology, I can mainly offer suggestions to modify the manuscript for improvement:

1) I do not understand the purpose of the chemical modification data presented on pages 14-15. I assume these data were included to strengthen the paper, but as it stands, I believe they do the opposite. A major finding of the study is that dsRBDs 1 and 2 bind equally well to both ends of an asymmetric siRNA duplex. This result leads to a very clear and important conclusion: TRBP is not a sensor of siRNA asymmetry. The chemical modification data clearly suggest asymmetry is at play in the knock-down experiments as "Chemical modification of the passenger strand residues yielded milder effects and a different profile from that obtained for the guide strand". Thus, I would naturally conclude that the chemical modification effects do NOT arise via perturbed interactions with TRBP. I therefore believe the statement, "we observe a clear correlation between how modifications in a number of residues of EL86 reduce translation repression and the presence of intermolecular contacts for these residues in our TRBP dsRBD12 structures" is not in keeping with the other findings in the study. If the authors want to make this claim they should observe interactions between TRBP12 and the modified siRNAs, as they did using EL86. However, it seems more likely to me that these modifications are impacting another step in the silencing process, such as interactions with Ago2, which have been shown to be sensitive to bulky 2' modifications in 5' end of the siRNA seed region (Schirle 2016, PMID: 27380263; Prakash 2005, PMID: 15974578).

We agree with referee #1 that the experimental support brought by our chemical modifications data to our structures is subject to caution. The main reason being the link between TRBP-EL86 interactions and the silencing inhibition triggered by chemical modification has not been demonstrated. The fact that the modifications' effects are observed only on the guide strand can also be misleading. Reporter data for a passenger strand should have also been included in order to truly support TRBP's symmetric binding.

We chose therefore to perform another experiment to support better our structures. Besides dsRBD12 symmetric binding, one of our main findings is that TRBP dsRBD12 binds on one face of the typical siRNA EL86 (the face presenting one major groove flanked by two minor grooves), leaving accessible the other siRNA's face (one minor groove flanked by two major grooves). This led us to propose that TRBP and Dicer binding surfaces on a pre-miRNA do not overlap, and therefore, that TRBP could bind on a pre-miRNA before / during its cleavage by Dicer. To support this hypothesis, we performed a Dicer-mediated cleavage assay using a pre-

miRNA derived from EL86 as a substrate. We then studied dsRBD12's impact therein, using different dsRBD12:RNA ratios. Essentially, we observe no dsRBD12 effect—or maybe a very small enhancement—for dsRBD12:RNA ratios between 0 and 5. A clear inhibition of the cleavage activity is observed at higher ratios. Considering that the RNA binding affinities of TRBP dsRBD12 and Dicer for the pre-miRNA are comparable [Chakravarthy et al., J Mol Biol, 2010], we conclude that there is no competition between these two proteins when binding the pre-miRNA at a 1 to 1 stoichiometric ratio, confirming that they bind distinct RNA regions, as hinted by our structure. We revised our manuscript to present and discuss these data in the Results and Discussion parts. The chemical modifications data are still shown and now better explained.

2)Figure 6: Although Giardia Dicer is the highest resolution Dicer structure currently available, this protein lacks the C-terminal dsRBD and helicase domains present in human Dicer. Therefore, the paucity of steric clashes in the presented TRBP-Dicer model is perhaps not so remarkable. Several moderate resolution EM structures of human Dicer (Lau 2012 PMID: 22426548; Taylor 2014, PMID: 23624860), as well as a structure of TRBP-dsRBD3 bound to a domain from the human Dicer helicase (Wilson 2014, PMID: 25557550), are available. Integrating the new TRBP12 model with these structures would be far more interesting and informative.

We prepared a new figure integrating more recent structural data. Our model now integrates the X-ray structures of Dicer's Platform-PAZ-Connector region [Tian et al., Mol Cell, 2010], RNase III domains [Du et al., Proc Natl Acad Sci U S A, 2008], Dicer-TRBP's interface [Wilson et al., Mol Cell, 2015], and the EM envelop of apo-Dicer [Taylor et al., Mol Cell, 2013]. We used this new model to discuss the possible occurrence of steric clashes between TRBP dsRBD12 and various Dicer domains (dsRBD, PAZ, RNase III). The distance TRBP dsRBD2-dsRBD3 was also compared with the length of the corresponding inter domain linker.

3)Discussion section: regarding guide strand selection the authors write of, "two main directions have been proposed to date..." with 1) thermodynamic asymmetry being detected before loading into Ago; and, 2) 5' nucleotide identity being used as a determinant by Ago upon loading. I do not believe this rigid dichotomy accurately reflects current thinking in the field. Noland 2013 (PMID: 23531496) suggest that Ago2 and Dicer-TRBP both contribute to guide selection, and Suzuki, 2015 (PMID: 26098316) suggest that Argonaute senses both thermodynamic asymmetry and 5' nucleotide identity. Therefore, my sense is that the general thought in the field is that there may be multiple steps that contribute to guide strand selection. The difficulty is determining the extent to which each putative step contributes, and the mechanisms underlying the different steps. I suggest rewording this part of the Discussion to provide a more holistic representation of the field.

The first part of the discussion—"Role of TRBP in mi/siRNA asymmetry"—has been substantially rewritten to present a more nuanced view of the current understanding of strand selection mechanisms.

4)Top of Page 4, "While the miRNA duplex is loaded into Ago, only the guide strand is retained..." consider including references to Leuschner PJ, 2006 (PMID: 16439995), and Martinez, 2006 (PMID: 12230974).

This reference has been included

5)Page 4, "While in flies the heterodimer Dicer-2/R2D2 was shown to be responsible for asymmetric RISC loading (Tomari, 2007)..." this statement should be qualified with a reference to Nishida, 2013 (PMID: 23375501), which showed that "R2D2 is not absolutely required for siRNA strand selection in vivo."

This reference has been included

6)Page 4, Suzuki, 2015 (PMID: 26098316) should be cited when discussing previous work towards understanding guide strand selection in human RISC in the Introduction and Discussion sections.

This reference has been included

7)Page 13, "the spatial arrangement of TRBP dsRBDs on EL86 primarily results from the recognition of the structural features of the A-form RNA helix..." How would binding be impacted by mismatches and bulges, commonly found in pre-miRNAs?

We agree this is an important question since many, if not most, pre-miRNAs have bulges or internal loops in their stem region. We chose, however, to focus our study on the recognition of a regular RNA stem by dsRBD12. We lack therefore the experimental data to answer this question. Nevertheless, our structure provides a detailed view of dsRBD12-RNA interactions that can be used to assess the impact of structural irregularities on dsRBD12 binding to RNA.

8)Page 4/5: It is implied that human TRBP is used, but never actually stated until the Method section on Page 18.

Human TRBP is now explicitly mentioned at the very beginning of the Results (p. 5).

9)Page 5: "we built structural models showing that TRBP has the potential to bind pre-miRNA prior [sic] Dicer cleavage". Neither human Dicer, nor a canonical pre-miRNA are used for the structural models, making this statement slightly inaccurate.

We modified the last paragraph of the Introduction (p. 5) to remove the reference to the modeling work. We refer instead to the results obtained from the Dicer-mediated cleavage assays.

10)The supplemental material lacks a methods section for the cellular assays, particularly with respect to the RT-qPCR (method, internal standard, calculations).

These missing data have now been added.

11)Fareh et al., 2016 is cited in the Discussion but does not appear in the References.

This article has been added to the References.

Referee #2

This is a solid and competently performed structural analysis of TRBP-siRNA recognition. The authors use a plethora of state-of-the-art biophysical strategies to measure molecular interactions, distances etc. resulting in highly relevant structural information (most importantly, the clear data towards sensing asymmetry in the duplex). However, the study lacks clearer in vivo validation experiments. Specific points are listed below.

1. The authors use a highly asymmetric siRNA and do not observe any effects on the orientation of the two dsRBDs suggesting that TRBP is not involved in strand selection. This is clearly and important information, which will be highly valuable to the field. Nevertheless, it would be nice to see this analyzed in a more systematic way. Is it possible to study a symmetric and a highly asymmetric siRNA in parallel? This would clearly experimentally strengthen the suggested model.

The referee points out an important issue, namely, whether dsRBD12's symmetric (or 'un-polarized') binding on the functionally asymmetric siRNA EL86 is generalizable to other RNA sequences (either functionally symmetric or asymmetric) or whether it is specific to EL86. We addressed this question by performing two additional single molecular FRET experiments with two other siRNA duplexes: the symmetric sod1 and the asymmetric pp-luc [Gredell et al., Biochemistry, 2010].

Our data clearly shows that TRBP dsRBD12 binds each of the pp-luc and sod1 siRNA in two symmetric orientations, as was already observed for the functionally asymmetric EL86. We therefore propose that TRBP dsRBD12's symmetric binding on siRNAs is independent of the RNA sequence, but is rather a general property of TRBP. We have updated the single-molecule FRET section, the discussion, and the Supporting Information accordingly.

2. $\alpha 0$ is suggested to help restricting the registers that can be used. This is suggested by the structure but not experimentally validated. Deletion of this helix could result in dramatically changed distance measurements, which would again experimentally validate the results.

We agree with Referee #2 that the hypothesis that helix $\alpha 0$ restricts the number of accessible binding sites is not validated experimentally. We have therefore removed this statement in the revised version.

3. The authors suggest that TRBP recognizes the 'shape' of the duplex siRNA. What does that exactly mean? Is the shape of a siRNA at the ends different than in the central region? Is there any data that could be used as reference for such a statement?

We actually stated in our abstract that TRBP's dsRBD12 recognizes EL86 'shape-specifically' rather than 'sequence specifically'. We replaced this statement with the following one: "We find that TRBP's dsRBDs recognize the structure of the siRNA's A-form helix rather than its nucleobase sequence.", which is more accurate.

In the last paragraph of the "Tertiary structure of the dsRBD12 – EL86 complex" section within the Results part, we indeed suggest that the arrangement of the individual dsRBD on EL86 is driven mainly by structural and dynamics variations occurring across the EL86 RNA helix. This is in line with a number of reports that demonstrate that dsRBDs recognize A-form RNA helices [Ryter et al., EMBO J, 1998; Vukovic et al., Biochemistry, 2014; Gong et al., Nat Struct Mol Biol, 2014]. The extremities of RNA (and DNA) double helix are more dynamic than the central region—a phenomenon called 'fraying' [Zgarbova et al., J Chem Theory Comput, 2014; Nonin et al., Biochemistry, 1995]. Furthermore we point out that the localization of a dsRBD's helix $\alpha 1$ and loop 2 at the end and at the center of an RNA helix, respectively, is reminiscent of a trend reported in other structural studies [Stefl et al., Cell, 2010; Wang et al., Structure, 2011].

4. The authors observe two different orientations in their structural model (complex A and B). Is it possible to include the structural information about the third dsRBD in complex with the helicase domain of Dicer (Wilson et al.)? Maybe new constraints appear that would favor one or the other complex? This should be analyzed and added if possible.

As stated in our answer to Referee #1's second comment, we have built a new model integrating more recent structural EM and X-ray data [Du et al., Proc Natl Acad Sci U S A, 2008; Wilson et al., Mol Cell, 2015; Taylor et al., Mol Cell, 2013]. In particular we have now included in our model the structure of TRBP dsRBD3 in complex with Dicer's PBD [Wilson et al., Mol Cell, 2015].

5. Figure 6A is not referenced in the text.

Figure 6A has been moved and included in Figure 5. It is now being properly referenced.

6. The validation data is not very strong. The authors even state in the results section that "...efficiency may stem from different potential causes...". Therefore, as presented, it does not contribute much. It would be clearer if for example Ago loading would be analyzed. In this case, at least target binding properties of the modified siRNA strands could be ignored.

As explained in more details in the answer to Referee #1's first comment, we share the view that our chemical modifications data do not support directly our dsRBD12-EL86 structures. Therefore, we decided to perform a Dicing assay with a pre-miRNA deriving from EL86. Our results show that dsRBD12 does not compete with Dicer (for dsRBD12:RNA ratios between 0 to 5), even though TRBP's affinity for pre-miRNA are comparable [Chakravarthy et al., J Mol Biol, 2010]. Therefore we propose that dsRBD12 and Dicer bind two distinct (i.e. non-overlapping) regions of the pre-miRNA up to a 1 to 1 stoichiometric ratio. We then built a structural model of a Dicer – RNA – dsRBD12 complex by superposing the RNA fragments present in each structure. Using this unbiased approach, we obtained a model where TRBP dsRBD12 and Dicer's RNaseIII and PAZ domains are positioned on opposite sides of EL86, in agreement with our proposal.

Referee #3

In this manuscript, Masliah et al. report the structure of the N-terminal two dsRBDs of TRBP (but lacking the 3rd dsRBD at the C-terminal) in complex with an siRNA duplex. The combination of NMR, EPR and single-molecule analysis nicely and convincingly shows the detailed tertiary structure of the dsRBD1,2-siRNA complex. However, the functional/biological insights that can be gained from the structure is rather limited and the discussions are often highly speculative. In my opinion, this manuscript is suitable for a more specific, structure-focused journal

Major points:

1. Previous studies have demonstrated that the main function of TRBP is to improve the accuracy of Dicer cleavage site on a subset of pre-miRNAs. Unfortunately, however, the current structure does not provide direct mechanistic explanation for this function of TRBP. Rather, all the related discussions are highly speculative based on the dsRBD1,2 structure (lacking the C-terminal dsRBD) with an siRNA duplex (not a pre-miRNA hairpin) and the docking model with Giardia Dicer (much smaller than human Dicer). Thus, the functional insights from the current structure are quite limited. Fig. 6 is too speculative and should at least be sent to Supplementary Information.

We agree with Referee #3's critics that our ternary structural model could be improved to give a more realistic picture of pre-miRNA cleavage by Dicer. We have therefore slightly reorganized our manuscript as follow:

- **In the first version of our manuscript we stressed one particular aspect of the dsRBD12-EL86 complex, which is the presence of two equally populated species related by a swapping of the individual domains. While this finding is important in the context of siRNA asymmetry 'sensing', we did not stress enough another aspect, which is the fact that dsRBD12 covers a limited area of EL86, leaving the remaining surface potentially accessible to other protein factors.**
- **Knowing that TRBP can influence pre-miRNA cleavage by Dicer, we made the hypothesis that during pre-miRNA cleavage, Dicer and TRBP bind the pre-miRNA simultaneously on two distinct (i.e. non-overlapping regions). To test this hypothesis, we designed a pre-miRNA deriving from EL86, which we used in a Dicer-mediated cleavage assay, performed in presence or in absence of dsRBD12. Observing no inhibiting effect from dsRBD12 (as we would expect if it were competing with Dicer to bind EL86), we conclude that these data support a model where dsRBD12 and Dicer bind simultaneously pre-miRNA during its cleavage by Dicer.**
- **Then we go on building a structural model with Dicer, dsRBD12 and dsRNA. Several X-ray structures show that *A. aeolicus*' RNase III domains recognize an RNA surface consisting of two major grooves separated by one minor groove. Looking at our structure, we see that the RNA surface covered by dsRBD12 consists of two minor grooves flanking one major groove, whereas the RNA surface left accessible presents the two major grooves / one minor groove required for RNase III binding. We then make the assumption that RNase III - dsRNA recognition is conserved in Dicer to build our structural model.**
- **We now conclude by saying that our ternary model represents a physically realistic picture as to how TRBP, Dicer and pre-miRNA can be brought together during pre-miRNA cleavage.**
- **Figure 6 has been modified to reflect this new organization. It also incorporates now more recent structural data, along with information on TRBP dsRBD3 and Dicer's own dsRBD. More details are to be found on the main text and in Figure 6.**

2. It is interesting that the two dsRBDs binds equally to the two ends of the siRNA duplex, in contrast to earlier studies reporting that TRBP can sense the thermodynamic asymmetry of small RNA duplexes. However, by using TRBP knockout cell lines, it has been demonstrated that the effect of TRBP on the guide strand selection is mostly indirect, via the above-mentioned shift of the Dicer cleavage site on a subset of pre-miRNAs (Kim et al., Cell Reports 2014). Thus, the functional importance of the binding mode between TRBP and an siRNA duplex is obscure (unlike the well-

established role of Dicer-2/R2D2 binding to an siRNA duplex in *Drosophila*). Again, this limits the biological insights that can be gained from the current structure.

While the report mentioned above (Kim et al., Cell Reports 2014) shows very convincingly that TRBP plays a role in pre-miRNA cleavage accuracy, which in turn drives miRNA arm switching, it does not investigate whether TRBP could still ‘sense’ RNA thermodynamic asymmetry. Direct asymmetry sensing by TRBP could have come into play in a different context such as during siRNA loading into Argonaute. We show, however, that this is not the case.

There are a few contradictory reports in the field, as whether TRBP plays a direct role in mi/siRNA asymmetry sensing or not, and our study clearly help clarifying the situation in that respect as pointed by the other referees. Furthermore as explained in the answer to the first comment, we have slightly shifted the focus of our article to discuss in more details the insights obtained for pre-miRNA processing.

Minor Points:

1. The authors only used a single siRNA duplex EL86. It will be safe to use at least one more highly asymmetrical siRNA duplex with completely different sequence before making a conclusion that dsRBD1,2 binds to siRNAs symmetrically. Also, direct comparison between NMR and biochemical analysis (i.e., crosslinking) using the same siRNA duplexes will be helpful to fill the gap between the current study and previous reports.

As detailed above (referee 2, remark 1), we performed additional single-molecule FRET experiments using a symmetric and a highly asymmetric siRNA. Our results clearly demonstrate that the orientation of the two dsRBDs is not affected by the RNA sequence, substantially strengthening a central conclusion of the manuscript. As these sequences have been used in earlier crosslinking studies [Gredell et al., Biochemistry, 2010], we were also able to directly compare our biophysical analysis with earlier biochemical experiments. We have updated the single-molecule FRET section, the discussion, and the Supporting Information accordingly.

2. Although single-molecule data are beautiful, the possibility that the introduction of fluorescence dyes affects the proper TRBP function cannot be excluded. Functional comparison between wild-type TRBP and dye-labeled TRBP should be conducted. The possibility that the 3'-end Cy3 modification of the siRNA affects TRBP binding should also be tested.

Incorporation of fluorophores has indeed previously been reported to lead to alterations in protein function. To rule out this issue, we performed RNA binding experiments to quantify the affinity of labeled TRBP towards EL86. We quantified the dissociation constant to be 210 ± 30 pM, a value that is consistent with earlier results obtained without fluorophores [Yamashita et al., Protein Sci, 2011]. Hence, proper TRBP function does not appear to be affected upon incorporation of fluorescent dyes.

The possibility that the 3'-end Cy3b modification of the siRNA affects TRBP binding was not tested, because the dye is conjugated to a 2-nt single-stranded overhang. However, since dsRBDs specifically recognize A-form RNA [Vukovic et al., Biochemistry 2014], it seems to be unlikely that a modification of the overhang results in changes in protein-dsRNA interaction. We have updated the single-molecule FRET section and the Supporting Information to clarify this point.

3. The data in Fig. S4 is problematic; there is no repetition and statistical analysis. In addition, the authors should note that guide strand modification may inhibit the siRNA function at step(s) other than TRBP binding, including Ago loading or RISC maturation.

Such analysis has now been performed.

4. The NMR data are somewhat difficult for non-specialists to interpret. More comprehensive explanation is recommended (e.g., in Fig. 3, how did the authors build 14 different models? How did they narrow down to two possible candidates indicated as green and magenta in Fig. 3B?).

A paragraph was added in the Methods to explain how the 14 models were built.

The four models shown in Figure 3B (two in green and two in magenta) are the only ones that agree with Residual Dipolar Coupling (RDC). This is presented in Figure 3A, which shows the r.m.s.d. between experimental and calculated RDCs (y-axis right) for each model (x-axis). We further narrowed it down to two models (the two green models, which are equivalent except for the swapping of dsRBD1 and dsRBD2), by measuring inter domain distances by Electron Paramagnetic Resonance (EPR).

We modified the text p. 9 and 10 to clarify this point.

2nd Editorial Decision

5th January 2018

Thank you for submitting a revised version of your manuscript to The EMBO Journal. It has now been seen by two of the original referees and their comments are shown below.

As you will see, the referees both appreciate the extensive amount of work that has gone into the revision and therefore support its publication here. However, ref #3 still raises a few concerns about the dicing assays that have been included in response to the reviewer comments. At this point, I will not ask you to redo all the experiments to address these points but in case you have data available for the effect of adding full-length TRBP, rather than the isolated RBDs, I would strongly encourage you to include it. In addition, please discuss your findings in the context of the NAR paper the reviewer mentions.

Given the overall positive input from the referees I would like to invite you to submit a final revision of the manuscript in which you address the reviewer comments (as outlined above) as well as the following editorial issues concerning text and figures:

-> Please reduce the number of keywords from 6 to 5.

-> Please indicate the nature of the error bars and the number of replicas used for calculating statistics for fig 6B.

-> Please include a callout for fig EV2. In addition, we noticed that there are callouts to fig S2C on p.7 and to fig S4C & D on p.21, but that these figures don't exist. Presumably, you meant to say fig S3C and D?

-> Please use the nomenclature 'Appendix Table S1-3' when referring to the Appendix tables in the main text (currently referred to as 'Supplementary Table S1-3')

-> For the figures, we noticed that panels A-C in Fig EV4 are not described in the figure legend. In addition, the text looks somewhat pixelated for some figures (eg fig 4) so I would encourage you to upload a version with a higher resolution, if possible.

Thank you again for giving us the chance to consider your manuscript for The EMBO Journal, I look forward to your revision.

REFEREE REPORTS

Referee #2:

In the revised version of their manuscript, Masliyah et al. have thoroughly addressed all points that I had raised on the previous version and included additional experiments and analyses.

One of my main concerns was whether or not the findings can be generalized since only one siRNA sequence had been used. The authors have addressed this by adding data on two additional siRNAs. A second concern was the validity of the structure model that had been presented. Now, additional available structures have been included and the model appears to be much more solid.

Therefore, I am satisfied with the revised version of the manuscript.

Referee #3:

There are two main conclusions in the revised manuscript: 1) Isolated dsRBDs1-2 of TRBP bind to an siRNA duplex in a symmetric manner and 2) TRBP's dsRBDs1-2 and Dicer's dsRBD simultaneously bind at the opposite sides of a pre-miRNA hairpin. The conclusion of 1) is well supported by the current data. However, a recent report (Tants et al., NAR 2017) has shown that Dicer can markedly enhance the asymmetric binding of full-length Loqs-PD (a TRBP homolog in flies) to an siRNA duplex. Given that this paper was published during the revision process, I don't think any further experiments are required but this point should be at least discussed carefully. 2) is more problematic. First of all, based on previous findings that TRBP enhances Dicer's activity, especially in RNA-crowded environments (Fareh et al., Nat Comm 2016), it was expected and unsurprising that dsRBDs of Dicer and TRBP do not compete. Unfortunately, the current study does not add much to our understanding of TRBP's role in pre-miRNA dicing, because the conclusion is based on the modeling using an siRNA duplex and lacks direct evidence; the pre-miRNA dicing assay (Fig. 6) was performed using the isolated dsRBD1-2, which (unlike full-length TRBP) cannot interact with Dicer. Overall, although I appreciate the authors' significant efforts to improve the manuscript and the technical soundness and thoroughness, the scope of this manuscript remains somewhat blurry.

Minor point:

As I as well as Reviewer #1 originally pointed out, 2'-MOE modifications may well affect other step(s) in RNA silencing such as Ago loading and RISC maturation, and the data in Fig. EV5 should be described more cautiously.

|

2nd Revision - authors' response

10th January 2018

Thank you for considering our revised manuscript for publication in the EMBO Journal. We have now edited our manuscript to address all editorial issues mentioned in your last email. Furthermore, we now discussed our results in the context of the recently published study by Tants et al., in more details as suggested. The following paragraph was included in the section "Role of TRBP in mi/siRNA asymmetry" of the discussion: "It has been reported in a very recent study that in flies, the heterodimer Loqs-PD-Dcr2 is able to discriminate the two ends of an asymmetric RNA. This ability relies on a moderate intrinsic binding preference of Loqs-PD for the most stable end of a 21-bp siRNA [Tants et al. 2017]. This apparent difference in the mode of binding of these functionally homologous proteins with regard to RNA asymmetry sensing is likely to be caused by a sequence divergence between the two proteins, in particular within the residues binding RNA and in the length of the inter domain linker, which is shorter in Loqs-PD. Additionally, the longer siRNA substrate used in the work by Tants et al. (two base-pairs longer) might also influence the mode of binding."

We haven't performed our dicing assay in presence of full-length TRBP, so we were unfortunately not able to follow your suggestion to include these data in our manuscript. However, the goal of our assay was to show that dsRBD12 and Dicer bound different regions of the pre-miRNA. In this context, we assume that the interaction Dicer-TRBP is not essential.

We hope that with these changes you will find the manuscript now suitable for publication in the EMBO Journal.

Accepted

12th January 2018

Thank you for submitting the final revision of your manuscript, I am pleased to inform you that the study is now officially accepted for publication in The EMBO Journal.